# Fine and coarse dust radiative impact during an intense Saharan dust outbreak over the Iberian Peninsula. Short-wave direct radiative effect

María-Ángeles López-Cayuela[1], Carmen Córdoba-Jabonero[1*], Michaël Sicard[2,#], Jesús Abril-Gago[3,4], Vanda Salgueiro[5,6], Adolfo Comerón[2], María José Granados-Muñoz[3,4], María João Costa[5,6], Constantino Muñoz-Porcar[2], Juan Antonio Bravo-Aranda[3,4], Daniele Bortoli[5,6], Alejandro Rodríguez-Gómez[2], Lucas Alados-Arboledas[3,4] and Juan Luis Guerrero-Rascado[3,4]

[1]Instituto Nacional de Técnica Aeroespacial (INTA), Atmospheric Research and Instrumentation Branch, Torrejón de Ardoz, 28850-Madrid, Spain
[2]CommSensLab, Dept. of Signal Theory and Communications, Universitat Politècnica de Catalunya (UPC), 08034-Barcelona, Spain.
[3]Andalusian Institute for Earth System Research (IISTA-CEAMA), 18006-Granada, Spain
[4]Department of Applied Physics, University of Granada (UGR), 18071-Granada, Spain
[5]Institute of Earth Sciences (ICT) and Earth Remote Sensing Laboratory (EaRSLab), Évora, Portugal
[6]Department of Physics, University of Évora, 7000-671, Évora, Portugal
[#]Now at: Laboratoire de l'Atmosphère et des Cyclones (LACy), Université de La Réunion, Saint Denis, France

*Correspondence to*: Carmen Córdoba-Jabonero (cordobajc@inta.es)

**Abstract.** Mineral dust has a key role in the Earth's radiative balance, and it has become significant over the Iberian Peninsula (IP), where Saharan dust outbreaks seem to increase in frequency and intensity. This study quantifies the dust direct radiative effect (DRE) in the short-wave range (SW) during an intense persistent springtime dust episode over the IP. Particularly, the vertical distribution of dust optical properties was derived at five lidar stations, and the Global Atmospheric Model (GAME) was used for radiative transfer simulations. Moreover, this study innovates by simulating the SW DRE using two distinct methodologies. The novel approach separates the impact of both fine (Df) and coarse (Dc) dust components and calculates the total DRE as their combined sum. In contrast, the commonly used one directly simulates the DRE for the total dust. Along the dust pathway crossing the IP, the SW DRE consistently registered a pattern of aerosol-induced cooling at both surface (BOA) and top-of-the-atmosphere (TOA). Results reveal that the Df role must be highlighted, as Df particles contributed nearly half of the total SW DRE at BOA and TOA, particularly for this event. DRE simulations based on the separation of dust modes for solar zenith angles < 70º show that nearly 75% out of DRE values are lower (in absolute value) with respect to those obtained by considering the total dust. That is, a less pronounced cooling effect is observed overall when the separation of dust modes is regarded, although relative differences between approaches are not highly significant in general (-3% and -6%, on average, at both BOA and TOA, respectively). However, this behaviour become opposite under moderate to high dusty conditions when the contribution of their Dc and Df components is separately considered, i.e., dust induces a more pronounced cooling effect. This indicates the relevant role of the fine dust particles in DRE estimations, likely underestimated when total dust (no separation) is taken into account in relatively high dusty environments. In addition, that cooling effect is more evident at TOA

than at BOA, probably due to the presence of dust at higher levels than those usual in the troposphere. This fact can introduce relevant implications for radiometric measurements from satellite instrumentation.

## 1. Introduction

According to the latest Intergovernmental Panel on Climate Change technical report (IPCC, 2023), not only is the Earth's global average temperature rising, but an increase in the frequency, intensity, and duration of extreme hot events and extreme precipitation has also been observed over several zones in the last few years. In this context of climate change, aerosols play a crucial role in climate forcing, both through their direct impact by interacting with radiation (aerosol-radiation interactions, ARI) and their indirect effect by serving as cloud condensation nuclei and ice nucleating particles (aerosol-cloud interactions, ACI). From 1750 to 2014, aerosols contributed to the global effective radiative forcing at the top of the atmosphere (TOA) by -1.3 W m$^{-2}$, resulting in a cooling effect (Arias et al., 2021), which partially compensates for the warming effect by increasing greenhouse gas emissions.

Among aerosol particles, mineral dust is one of the most abundant and widely distributed in the Earth's atmosphere. Significant uncertainties remain in the estimation of the dust radiative effect, mostly due to the lack of observational constraints. Regarding the SW range, the balance between scattering and absorption is determined by the dust particle size and mineralogy, and the underlying surface's albedo determines the extent to which both processes impact the TOA radiative flux (Kok et al., 2023). In particular, the Sahara desert and Sahel regions are the major contributors to mineral dust lifting, accounting for over 50 % of global emissions (Kok et al., 2021). On one hand, the geographical proximity of the Iberian Peninsula (IP) to North Africa, as well as the persistence of certain favourable weather patterns (Russo et al., 2020; Couto et al., 2021), makes the IP one of the main pathway regions for Saharan desert dust transport towards Europe. On the other hand, Sánchez-Benítez et al. (2020) found that Iberian heatwave events are primarily linked to anomalous atmospheric circulation, typically characterized by a pronounced positive 500 hPa geopotential height anomaly (Z500) aloft. In particular, an analysis of daily weather regimes during Iberian heatwave days reveals a marked dominance of positive Z500 anomalies over Western Europe, resembling the occurrence of Euro-Atlantic subtropical ridges. The frequency of this occurrence during heatwave days is observed to double relative to climatological averages. Last studies linked both processes, showing robust evidence for increases in maximum temperatures and the frequency of heatwaves over Europe (IPCC, 2023), which can be partially associated with dust intrusions (Sousa et al., 2019; Fernandes and Fragoso, 2021; Barriopedro et al., 2023). While there are certain indications that future global dust aerosol concentrations may increase (Allen et al., 2016; Tegen and Schepanski, 2018), this likelihood is notably contingent upon alterations in precipitation patterns and atmospheric circulation (IPCC, 2023). Consequently, the radiative consequences of such potential changes remain unclear (Allen et al., 2016; Kok et al., 2018). Since pre-industrial times, the dust mass that has lifted into the atmosphere in those North African regions has increased by 46% (Kok et al., 2023). Moreover, there is not only an increasing frequency of Saharan dust outbreaks over the IP, when compared with long-term records (Sousa et al., 2019; Salvador et al., 2022), but also in the number of studies reporting extreme and intense dust episodes in the region

(e.g. Guerrero-Rascado et al., 2009; Preißler et al., 2011; Santos et al., 2013; Cazorla et al., 2017; Sorribas et al., 2017; Titos et al., 2017; Valenzuela et al., 2017; Córdoba-Jabonero et al., 2019, 2021; Fernández et al., 2019; López-Cayuela et al., 2023).

This study focuses on the interaction of Saharan dust with radiation, i.e., on the direct radiative effect (DRE) of dust particles. By using observational measurements, the estimation of the DRE is performed, underscoring the importance of determining their radiative properties and accurately quantifying their direct radiative impact on the Earth-atmosphere radiative budget. This work aims to study the DRE during an exceptionally intense and long-lasting (14 days, from 25 March to 7 April 2021) Saharan dust event over the IP, allowing the study of changes in the radiative properties across both temporal and spatial

dimensions. This event was well characterized in the work of López-Cayuela et al. (2023) by determining the vertical behaviour of the optical and microphysical dust properties and using data obtained from five Iberian lidar stations. This study focuses on the direct radiative effect of the dust particles in the short-wave (SW) spectral range.

The use of lidar measurements should be emphasised as an added value of this study. Thanks to the application of the POLIPHON algorithm (Polarisation Lidar photometer Networking method; Mamouri and Ansmann, 2014, 2017; Ansmann et

al., 2019) to the polarized lidar measurements, it is possible to separate the fine and coarse dust contributions (Df and Dc, respectively) to the total DRE. In this work, the Df particle role has been highlighted in contrast to other studies, in which the potential impact of Df particles on the total radiative effect has not been separately underlined. The reason for overlooked Df contribution is that dust intrusions are typically dominated by large particles (Dc), in particular, near the source. Indeed, the DRE of total dust (DD) has been extensively studied in both SW and LW spectral ranges (e.g. Sicard et al., 2014a, 2016;

Barragán et al., 2016; Granados-Muñoz et al., 2019) but only a few studies have addressed this issue by separating both dust components. For instance, Sicard et al. (2014b) reported that Dc particles primarily affect LW solar radiation, while Df is primarily responsible for SW radiative modulation. Furthermore, the impact of Df particles in the SW spectral range of the solar radiation is greater at the top-of-atmosphere (TOA) than at the surface (BOA) (Córdoba-Jabonero et al., 2021), while the contribution of the Dc particles in the LW solar spectral range at the TOA is lesser than at the BOA (Sicard et al., 2022).

Hence, it is essential to study both Df and Dc particles separately when estimating individual radiative effects.

It should be highlighted that in dust climate models, the separation can be particularly helpful for analysing DRE per size mode. On one hand, when dust is treated as a singular entity (total dust), the contribution of fine dust can be masked, as the radiative properties input into radiative transfer models tend to be skewed towards the characteristics of coarse dust. On the other hand, recent studies have found evidence that Dc is inadequately represented in global atmospheric models (Adebiyi and

Kok, 2020). In particular, the mass of coarse dust in the atmosphere could be about four times greater than simulated in current AeroCom climate models (https://aerocom.met.no/). Since Dc warms by absorbing both SW and LW radiation (Kok et al., 2017), the underestimation of Dc by both climate models indicate that the net DRE could be more warming than has been previously estimated. Overall, the atmosphere appears to contain approximately 40% more dust (both Df and Dc) than what is simulated by AeroCom models, accounting for about 80% of the total particulate mass load in the atmosphere (Textor et al.,

2007; Adebiyi and Kok, 2020). Thus, Df and Dc distinction is particularly relevant for data assimilation processes in climate models, where accurately representing the radiative properties of different dust components is critical for improving model

performance and predictions. As an added value, this study introduces the novelty of simulating SW dust DRE using two distinct approaches. On the one hand, by simulating the contribution of Df and Dc components separately, and then estimating the total dust DRE as their sum (DD = Df + Dc). On the other hand, the DRE is directly simulated for the total dust component.

The paper is organized as follows. The GAME radiative transfer model and the parametrizations used in terms of SW range are described in Section 2. An overview of the dust outbreak can be found in Section 3. In Section 4 the results and discussion are exposed. Finally, the main conclusions of this study are found in Section 5.

## 2. Methodology

### 2.1 Radiative transfer model: GAME

GAME (Global Atmospheric Model) is the radiative transfer model (RTM) used in this study for computing the solar radiation fluxes. Particularly in this work, they are simulated in the SW spectral range (0.2 to 4.0 μm, with a wavenumber resolution of 400 cm$^{-1}$ from 0.2 to 0.7 μm and 100 cm$^{-1}$ from 0.7 to 4.0 μm). Hereafter, for simplicity, the 'SW' notation is omitted. GAME was developed by Dubuisson et al. (1996, 2004) and employs the discrete ordinates method (DISORT; Stamnes et al., 1988) to compute the solar flux at the boundary of homogenous and plane atmospheric layers. The SW radiation flux is calculated for 18 layers of the atmosphere ranging from ground level to an altitude of 20 km height.

This RTM accounts for the absorption of gases, specifically $H_2O$, $CO_2$, $O_3$, $N_2O$, CO, $CH_4$, and $N_2$, using the correlated k distribution (Lacis and Oinas, 1991). Additional information on the calculation of the gas transmission functions can be found in Dubuisson et al. (2004) and Sicard et al. (2014b). The parameterization of gas absorption is based on pressure, temperature, and relative humidity profiles. Particularly, the data from the Global Data Assimilation System (GDAS), provided by the National Oceanic and Atmospheric Administration (NOAA) are used (ftp.arl.noaa.gov/archives/gdas1/; last access: 28 February 2023). Additionally, the model requires input from the spectrally resolved surface albedo (SA), whose data are obtained from MODIS (Moderate Resolution Imaging Spectrometer; https://modis.gsfc.nasa.gov; last access: 28 March 2023). The input parameters and radiative properties as introduced in the GAME model for the SW spectral range and the databases used are summarized in Table 1.

### 2.1.1 AERONET properties used as input in GAME

In GAME model, aerosols are included by comprehensively parameterizing their radiative properties in terms of the spectral dust extinction coefficient ($\alpha$), single scattering albedo ($\omega$), and asymmetry factor ($g$). This section focuses on both $\omega$ and $g$ taken from NASA's AERONET (AErosol RObotic NETwork; http://aeronet.gsfc.nasa.gov) Version 3 Level 2.0 data inversion products (last access: 18 February 2023). At the time of writing this study, fewer or no AERONET Level 2.0 data were available for all the stations. In such cases, AERONET Level 1.5 data were used instead. Those AERONET $g$ and $\omega$ are given with an uncertainty of ±0.03 and ±0.04, respectively, for dust particles (Dubovik et al., 2000, 2002).

Both $\omega$ and $g$ values are assumed to be vertically constant, and equal to the columnar values of $\omega$ and $g$. In addition, they are interpolated at the model wavelengths up to the largest SW wavelength provided by AERONET (1020 nm) and assumed to be constant beyond. In addition, they are hourly averaged, and interpolated in case those hourly values are missing. Also note that
$g$ is given separately for the fine, coarse and total modes.

## 2.1.2 Lidar-derived extinction used as input in GAME

Since the focus of this work is to obtain the DRE while considering the separation between Df and Dc, continuous hourly-averaged lidar measurements together with POLIPHON (Mamouri and Ansmann, 2017; Córdoba-Jabonero et al., 2018) method were applied to obtain the vertical profiles of the dust extinction coefficient at 532 nm ($\alpha^{532}$), instead of using the
aerosol optical depth given by AERONET. The POLIPHON approach relies entirely on empirically obtained dust input parameters, specifically the dust depolarization ratio and dust lidar ratio, without the need for a particle shape model. This method has undergone extensive validation and applied in several works (Ansmann et al., 2017; Mamouri and Ansmann, 2017; Córdoba-Jabonero et al., 2018, 2021; Couto et al., 2021; Salgueiro et al., 2021; Sicard et al., 2022). In the first step of POLIPHON, dust particles are separated from the remaining aerosol particles. In the second step, Df and Dc are identified,
enhancing the analysis of the backscatter profile. Once separated by particle modes, $\alpha^{532}$ for each component is obtained by considering the specific particle lidar ratio at 532 nm for each one (Ansmann et al., 2019). The uncertainties in the calculation of $\alpha^{532}$ using this methodology are 30-50%, 20-30%, and 15-25%, for Df, Dc and total dust, respectively (Ansmann et al., 2019).

The coarse and fine lidar-derived $\alpha^{532}$ profiles were previously obtained in López-Cayuela et al. (2023) (see Sect. 3). As those
profiles have different vertical resolutions depending on the lidar system used, they have been degraded to the 18 model layers (ranging from the surface up to 20 km height) through trapezoidal numerical integration in order to homogenize all the datasets. In addition, the lidar-derived $\alpha^{532}$ profiles are used together with the AERONET Ångström exponent at 440-870 nm (AE[440-870]) to derive the height-resolved dust extinction at other wavelengths within the SW range. Finally, hourly dust optical depth (DOD, $\tau$) values for both modes and for the total dust (DD) are computed by vertically-integrating those spectrally-estimated
dust extinction coefficient profiles.

## 2.2 Dust radiative effect estimation

The aerosol direct radiative effect (ARE) is particularly obtained for dust as the dust direct radiative effect (DRE), which can be defined at a given height ($z$) as follows (Córdoba-Jabonero et al., 2021; Sicard et al., 2022),

$$DRE(z) = \left[F_d^\downarrow(z) - F_d^\uparrow(z)\right] - \left[F_0^\downarrow(z) - F_0^\uparrow(z)\right], \tag{1}$$

where $F_d$ and $F_0$ denote the solar radiative flux (W m$^{-2}$) as computed by GAME, respectively, with (d) and without (0) dust occurrence. The ↓ and ↑ arrows indicate whether the fluxes are downward or upward, respectively. The first ($[F_d^\downarrow(z) - F_d^\uparrow(z)]$) and second ($[F_0^\downarrow(z) - F_0^\uparrow(z)]$) terms of Equation (1) represent, respectively, the net flux with and without dust presence.

According to that definition, a negative/positive DRE value represent a cooling/warming effect. In this study, the DRE is shown at both TOA and BOA. The dust contribution in the overall atmospheric column is evaluated by quantifying the atmospheric radiative effect, DRE(ATM), which is defined as follows,

$$DRE(ATM) = DRE(TOA) - DRE(BOA), \tag{2}$$

Note that $\alpha$ profiles are obtained from 'hourly' lidar measurements, and the DRE values denote 'instantaneous' simulated radiative values as obtained from those hourly $\alpha$ profiles; hence, for simplicity, both terms are removed. The fine-to-total ratio (Df/DD) of DRE (ftr_DRE) is also calculated, together with a linear fitting analysis of that variable over time. The slope of this linear fitting, denoted as $\delta DRE$, represents a measure of the temporal rate of the relative Df contribution to the DRE. Besides, the daily-averaged DRE is computed as follows:

$$DRE^{daily} = \frac{N_d}{24} \frac{\sum_{i=1}^{N} DRE^{hourly}}{N}, \tag{3}$$

where N is the number of hourly DRE values as computed for each day, and $N_d$ is the total number of hours with solar zenith angles (SZA) < 90° along the day (i.e. $N_d$ = 12 in this work). Moreover, the episode-averaged DRE is computed as the average of the daily-averaged DRE values for the entire event. For simplicity, hourly DRE will be denoted by DRE hereafter, if unless otherwise specified.

The dust radiative efficiency (DREff) is defined as the ratio of the DRE to DOD$^{532}$ values (in W m$^{-2}$ $\tau^{-1}$) along the day. Hence, the daily DREff is obtained from the slope of their linear fitting of DRE values with respect to DOD$^{532}$, forced to zero for each day.

## 2.3 Comparison between different approaches

The dust-induced DRE is calculated using two different approaches:

1) the previously exposed one, i.e., considering the contribution of the Df and Dc particles separately in the DRE, following the expression below:

$$DRE^{(I)} = DRE^{DD} = DRE^{Df} + DRE^{Dc}, \tag{4}$$

2) DRE is obtained from the contribution of the total dust (no separation),

$$DRE^{(II)} = DRE^{total}. \tag{5}$$

That notation will be employed throughout the entire paper. That is, 'DD' will refer to variables obtained by summing Df and Dc components, and 'total dust' will denote variables obtained without considering the mode separation.

The main distinction between the two approaches lies in certain radiative inputs used in the simulations. Specifically, in the
first approach ($DRE^{(I)}$), both $g$ and $\alpha$ properties are introduced as separate fine and coarse components, whereas in the second approach ($DRE^{(II)}$), these variables are introduced as their total component. All other variables are consistent for both approaches. Further details can be found in Table 1. Note that the second approach ($DRE^{(II)}$) is the most used in the literature. However, the novelty of this study lies in showing that the first approach ($DRE^{(I)}$) enables the dust DRE computation by using the optical properties of each dust mode separately (Sicard et al., 2014b; Córdoba-Jabonero et al., 2021; Sicard et al., 2022)
assuming that the dust-induced diffuse radiation from one of the modes does not interact with the other one. Moreover, it should be considered that the uncertainties in $DRE^{(I)}$ should be higher than in $DRE^{(II)}$, as derived from the uncertainties in the calculation of the POLIPHON $\alpha^{532}$, which are higher for Df and Dc modes than for total dust (see Sect. 2.1.2).

Finally, the differences found in the dust-induced DRE ($\Delta DRE$), and the relative differences ($\Delta^{rel}DRE$), between the two approaches can be calculated using the following equations:

$$\Delta DRE = DRE^{(I)} - DRE^{(II)}, \tag{6}$$

$$\Delta^{rel}DRE \ (\%) = 100 \frac{(DRE^{(I)} - DRE^{(II)})}{DRE^{(II)}}. \tag{7}$$

Since the dust SW DRE is negative, negative (positive) $\Delta DRE$ ($\Delta^{rel}DRE$) values indicate that $DRE^{(I)}$ exhibits higher absolute values than $DRE^{(II)}$. This suggests that a more pronounced cooling effect would be induced by dust according to the DRE simulations derived from the dust-mode separation approach ($DRE^{(I)}$) with respect to the total dust approach ($DRE^{(II)}$). Additionally, a statistical analysis based on the $\Delta^{rel}DRE$ percentiles ($P$) will be considered to assess the significance of discrepancies between both methodologies. In particular, the first ($P25$) and third ($P75$) quartiles will be examined.

**3. Overlook of the dust outbreak crossing the Iberian Peninsula**

During spring 2021, the IP experienced a significant and prolonged dust intrusion (25 March – 7 April). This event was closely monitored and analysed by five stations strategically located across the IP, namely El Arenosillo/Huelva (ARN), Granada (GRA), Torrejón/Madrid (TRJ), and Barcelona (BCN) in Spain, and Évora (EVO) in Portugal. Figure 1 shows the geographical position of each station in the IP. All these stations share the commonality of being dedicated to aerosol-cloud monitoring.

Among several instrumentation, each station is equipped with an AERONET photometer (or is close to an AERONET station) and a lidar system. Those five stations share a common exposure to Saharan dust outbreaks, particularly during spring and summer months, albeit with varying frequencies (i.e., Córdoba-Jabonero et al., 2021; López-Cayuela et al., 2023). Moreover, each station exhibits a unique aerosol background. Particularly, the aerosol background at the ARN station is mostly from marine and rural origin, as ARN is placed in a rural environment at the southwestern IP, and less than 1 km from the Atlantic coastline. EVO station is located in a rural region with limited industrialization and low levels of anthropogenic aerosol concentrations (Pereira et al., 2009; Preissler et al., 2013). Both GRA and TRJ stations are located in populated cities, and their background aerosols are of anthropogenic origin (Lyamani et al., 2012; Molero et al., 2014). Finally, BCN station is located on the northeast coast of the IP, within a densely populated and industrialized region, being thus the background aerosol load predominantly composed by urban and marine aerosols (Sicard et al., 2011).

Focusing on the dust intrusion studied in this work, a deep analysis of the synoptic situation and back-trajectory analysis revealed that the dust intrusion originated in the Saharan region and traversed the IP from the southwest to the northeast. The dust event was accompanied by an extended cloud cover across all five Iberian lidar stations, resulting in the unavailability of some lidar retrievals. Consequently, the dataset exhibits certain gaps, along with periods of no lidar measurements.

The extensive study of the dust outbreak can be found in López-Cayuela et al. (2023), together with the methodology to obtain the optical and mass properties derived from the polarized lidar measurements. This section aims to summarise the most significant results for the optical properties, which are essential for the development and discussion of the current study.

This dust event is particularly noteworthy not only due to the intensity of the dust outbreak but also because the prevailing meteorological situation conditioned the main findings. Among those, there were: 1) the detection of dust at upper layers of the troposphere in the northern half of the IP (reaching up to 10 km height), caused by the atmospheric instability that induced vertical motions of the air parcels, and 2) the absence of uniform gravitational settling observed throughout the IP, that is the Df/DD proportion remained nearly constant along the dust pathway across the IP (López-Cayuela et al., 2023), which contrasts with the long-range dust events observed in central Europe.

An overview of the temporal evolution of the dust outbreak crossing the IP in terms of the extinction ($\alpha$, km$^{-1}$) over the five Iberian lidar stations is shown in Figure 2. Particularly, the hourly-mean DD $\alpha$ at 532 nm ($\alpha_{DD}^{532}$) are shown. Despite some data gaps due to either no inversion possible or no lidar measurements, the transport of dust particles can be appreciated by looking at the behaviour of $\alpha_{DD}^{532}$ along the south-west IP (Fig. 2e) to the north-east IP (Fig. 2a) pathway of the dust intrusion during the overall dusty period (25 March – 7 April 2021). In terms of the episode-averaging, DD DOD$^{532}$ decreases as latitude increases, particularly ranging from 0.34 at ARN to 0.14 at BCN, without significant differences in the Df/DD ratio (ftr_DOD) between the stations (approximately 30%, see Table 2).

In this work, the hourly-averaged DOD$^{532}$ values are considered low when DOD$^{532}$ < 0.20, moderately intense when $0.2 \leq$ DOD$^{532}$ < 0.50, intense when $0.50 \leq$ DOD$^{532}$ < 1.00 and extreme for DOD$^{532} \geq 1.00$. Thus, this episode spans the full range, highlighting high and extreme DD DOD$^{532}$ values on the days of maximal incidence (29 March – 1 April). The mean and maximum hourly-averaged DOD$^{532}$ values for Df, Dc and DD components during the entire episode are shown in Table 2.

**4. Results and discussion**

This section is divided into an analysis of the specific radiative dust properties introduced in the model (Sect. 4.1), the DRE as estimated considering approach 1 (Sect. 4.2), and the discrepancies found by estimating the DRE from the two considered approaches (Sect. 4.3). GAME calculations are performed when the Sun is above the horizon (SZA < 90°). The DRE is calculated by using Equations (1) and (2), separately for the Df, Dc and total dust components (see Sect. 2.2).

**4.1 Radiative properties during the dust episode**

**4.1.1 Surface albedo (SA)**

For each station considered in this study, the SA values corresponding to the dust episode reveal a symmetric distribution (regardless of the wavelength under consideration). In fact, for each wavelength, the relative differences in SA throughout the episode at the same station are less than 0.1%. For this reason, although hourly values are considered for the simulations, Figure 3a represents the mean value of SA per wavelength and station.

However, and regarding the SA values (Fig 3a), a similar pattern across all sites is found, with SA increasing with wavelength until 870 nm, reaching the maximum values at that wavelength. Within the visible range (VIS, wavelengths < 780 nm) the values increase with latitude, showing $SA^{480} = 0.04$ at ARN and $SA^{480} = 0.07$ at BCN. This characteristic is not preserved at the Near-Infrared range (NIR, 780-2500 nm). EVO and TRJ show maximum SA values of 0.33-0.36 at 870 nm, which ranges from 0.26 to 0.20 for the other stations. Along the rest of the NIR range, the SA values decrease until $SA_{max}^{2130} = 0.13$-0.18.

Notably, the SA values in this study align with those found in Granados-Muñoz et al. (2019) and Córdoba-Jabonero et al. (2021) for GRA and BCN, respectively, in summertime.

Larger SA values indicate that, at a given wavelength and constant incoming radiation reaching the surface, more radiation will be reflected upward. Overall, minimum SA values are found in BCN, while maximum ones are found at TRJ and EVO stations. The differences between the five lidar stations are lower than 5% in the VIS range and for wavelengths greater than

1650 nm. At the beginning of the NIR range (870-1240 nm), EVO and TRJ reflect 7-15% more radiation than the other stations. As exposed in Sect. 3, the stations are located in a variety of places, from rural sites with different types of vegetation to urban sites. Thus, variations in SA values between stations are expected, as SA depends on the surface coverage.

**4.1.2 Asymmetry factor (*g*) and single scattering albedo (*ω*)**

Conducting the same analysis for the parameters *g* and *ω*, a skewed distribution is obtained for each station and wavelength,

with maximum relative differences during the dust episode of 1.5 and 2.5%, respectively. For this reason, and although once again the hourly-averaged value of these variables is used for each simulation, Figures 3b-3c depict an hourly data snapshot from the days of highest incidence, i.e., 1 April at 10 UTC for BCN station and 31 March at 12 UTC for the rest of the sites. The *g* and *ω* histograms for the five lidar stations can be found in the Supplementary Material (Figs. S1-S5).

The $g$ values decrease with the wavelength, being the forward scattering approximately 1.3 times greater for the coarse mode (coarse $g^{440} = 0.82\text{-}0.86$) than for the fine mode (fine $g^{440} = 0.67\text{-}0.70$). As explained in Córdoba-Jabonero et al. (2021), this result implies that the solar radiation scattered to the surface is greater for the coarse mode than for the fine mode, keeping a constant AOD and at low SZA, independently of the wavelength. Regarding the total $g$, its values range between those for the fine and coarse modes (total $g^{440} = 0.75\text{-}0.80$). The variability of $\omega$ is also low between the stations, increasing with wavelength Particularly, $\omega^{440}$ ranges from 0.90 to 0.93, being representative of weak absorbing particles. From 675 nm to NIR, $\omega$ increases until values of 0.98-0.99 for all stations.

Figure 4 shows the temporal evolution of $g^{440}$ for the fine and coarse components, together with both $g^{440}$ and $\omega^{440}$ for the total component at the five lidar stations. Along the entire dust outbreak, these properties remain relatively constant at each station. The variability observed is minimal, with values below 5 and 3% for both $g^{440}$ and $\omega^{440}$, respectively. It should be noted that, in this study, the variability has been quantified as the percentage ratio between the standard deviation and the mean value along the entire episode for each station. Specific values are detailed in Table 3.

As aforementioned, the coarse component of $g$ presents the most influence on scattered radiation. The lowest values of this parameter were observed in ARN and BCN, while the highest values are evident in GRA stations. Thus, the solar radiation scattered to the surface is lower for ARN and BCN than for GRA stations. Conversely, the pattern is reversed for $\omega$. However, it can be considered that the differences among the stations are marginal, displaying relative differences in the episode-average values of less than 3% for the total component in both $g^{440}$ and $\omega^{440}$. Regarding the separation of the dust components, the differences in the episode-averaged $g^{440}$ were 1.5 and 4% for Df and Dc, respectively.

In summary, the results for $g$ and $\omega$ are similar for all the stations, showing typical values expected for dust (Dubovik et al., 2002). Indeed, the differences between the stations are less than 5%, regardless of the variable and wavelength being examined. Thus, the differences expected in DRE between stations should mostly rely on the specific values of SA for each station (Sect. 4.1.1), and the variability of dust extinction between the stations and during the dusty episode. A summary of the evolution of the dust extinction can be found in Section 3, and a complete analysis can be found in López-Cayuela et al., (2023).

## 4.2 Dust-induced direct radiative effect

This section is devoted to describing the episode in terms of the evolution of the dust properties and DRE across the stations considered. Figure 5 shows the temporal evolution of the DRE and the ftr_DRE, at ARN station as an example. To avoid an excessively lengthy paper, the graphics for the rest of the stations can be found in the Supplementary Material (Figs. S6-S9). Figure 6-left shows DRE as function of DOD$^{532}$ at ARN, for instance, and Figure 6-right the DREff values as obtained for all the five lidar stations. Moreover, the specific episode-averaged and maximum DRE values together with DREff, ftr_DRE and $\delta DRE$ for each station, are shown in Table 4.

**4.2.1 DRE at BOA**

The DRE is negative at all stations, during the entire dusty period, indicating dust-induced cooling (see Fig. 5a and S6a-S9a, in purple). The intensity of the episode in terms of the radiative forcing is directly correlated with the DOD[532], being greater for Dc compared to Df particles. Thus, during the first part of the episode (until 1 April) the dust event was moderate, showing hourly Df (Dc) DRE values above -10 W m$^{-2}$ (-20 W m$^{-2}$). During the rest of the episode, the dust event is weaker, showing hourly DRE values 50% lower. On some specific days (27 and 31 March), the event is notably intense at ARN and TRJ stations

(mean AOD ~ 0.80), reaching hourly Df (Dc) DRE values below -40 and -20 W m$^{-2}$ (-60 and -40 W m$^{-2}$). Indeed, the maximum hourly DRE for the dust episode is found in ARN (DOD[532] = 1.30), while the rest of the stations exhibit maximum values of approximately 30-50% lower (Table 4).

The intensity of the overall dust episode can be understood by averaging the daily DRE values (episode-averaged DRE, see Sect. 3.2). The highest episode-averaged DRE values for DD are found at the southern and central stations, decreasing 40% at

EVO and 50% at BCN (see Table 2 and 4), with differences mainly due to the decrease in DOD with latitude. As expected, the differences found between stations are mainly related to the DOD decrease with latitude (see Table 2 and 4). Indeed, Meloni et al. (2005) found that the radiative forcing at both the TOA and BOA varies almost linearly with AOD, and the strong dependence of the DRE on AOD is well documented (Prasad et al., 2007; Sicard et al., 2014b; Lolli et al., 2018; Meloni et al., 2018; Granados-Muñoz et al., 2019).

DRE values are compared with other authors' findings over the IP. Córdoba-Jabonero et al. (2021) studied a moderately intense dust episode over BCN (maximum hourly-averaged DOD[532] of 0.60), by separating the Df and Dc components. The daily-averaged DRE results are similar for similar daily-averaged DOD[532] conditions. By considering other studies when the Df and Dc separation was not applied, for similar intense dust aerosol load conditions the daily total DRE reported on those studies are almost 50% greater than in the present study (Cachorro et al. 2008; Valenzuela et al., 2017). However, similar results are

found for extreme dust aerosol conditions (Bazo et al. 2023).

This study also analyses the DREff for Df, Dc and DD particles. The results show an excellent degree of confidence, with a correlation coefficient greater than 0.80 in all stations. By averaging those DREff values for the five lidar stations, a mean value of -138.5 ± 20.3 W m$^{-2}$ $\tau^{-1}$ (-95.3 ± 5.5 W m$^{-2}$ $\tau^{-1}$) is obtained for Df (Dc) particles. However, by considering the stations separately, the DREff slightly increases with latitude (i.e., from ARN to BCN) up to a maximum of 10% for all particle modes

(see Table 4; Figure 6). GRA station deserves special mention, located at a similar latitude as ARN and with similar dust optical properties (see Fig. 3) and Df/DD proportions, shows 5-10% lower DREff values for Df and Dc particles. This difference may be due to the limited data available in the study, resulting in a smaller range of DODs. Comparing our results with other studies over the IP, similar results are found, with maximum differences in DREff of 10% (Obregón et al., 2015; Sicard et al., 2016; Granados-Muñoz et al., 2019; Córdoba-Jabonero et al., 2021).

Finally, and regarding the contribution of the Df particles, the mean daily ftr_DRE at five lidar stations is around 40%, indicating that Df particles contribute almost half of the total DRE at BOA. Although there is a slight increase in $\delta DRE$ during the dust outbreak period, it remains below +1% day$^{-1}$ (Fig. 5c; Table 4).

### 4.2.2 DRE at TOA and in the atmosphere

This study shows that the DRE at TOA is negative at all stations, indicating a dust-induced cooling effect (see Figs. 5 and S6-
S9). Similarly to results found at BOA, the first part of the dust outbreak was moderate in terms of the radiative forcing (Córdoba-Jabonero et al., 2021) and weaker for the rest of the episode, showing slightly lower values than at BOA. Once again, the greatest hourly DRE values were found at ARN and TRJ on 27 and 31 March (see Figs. 5 and S6; Table 5). Overall, the stations exhibit maximum daily DRE values of approximately 35-70% lower than ARN. Newly, the highest episode-averaged DRE values for DD are found at the southern and central stations, decreasing 50% at EVO and BCN (see Table 5). In
comparison with other similar studies in similar dust load conditions over the IP, similar daily-averaged DRE results are found for Df and Dc particles (Córdoba-Jabonero et al., 2021), but 20%-30% lower for DD particles (Cachorro et al., 2008; Valenzuela et al., 2017).

The linear regression analysis shows a slight positive trend in the contribution of Df particles to total dust DRE during dust outbreaks, with higher $\delta DRE$ values at TRJ. However, GRA station shows negative values of almost 3% day$^{-1}$ (see Table 5).
The episode-averaged ftr_DRE shows similar values of approximately 45% at all lidar stations, suggesting that Df particles can induce almost half of the total dust DRE at TOA.

As for the BOA analysis, that linear fitting between DOD and DRE shows a good degree of confidence, with a correlation coefficient greater than 0.80. By averaging those values for all the five stations, a mean DREff of -107.1 ± 10.4 W m$^{-2}$ $\tau^{-1}$ (-57.5 ± 3.3 W m$^{-2}$ $\tau^{-1}$) for Df (Dc) particles are found, highlighting the higher variability of Df compared to Dc particles,
indicated by the standard deviation. The study of DREff allows assessing the radiative effect by eliminating the aerosol burden variable. However, not only is the aerosol load significant of DRE calculations, but also the structure and altitude at which the dust is located. The most notable difference among the studies from the literature is the dust layering. In this work, the dust particles reached several times the higher troposphere (over 6 km above sea level) over certain stations (TRJ and BCN). Thus, DRE is more pronounced and DREff is greater at DOD constant than a dust layer located at lower altitudes. For this reason,
our results could differ from other studies, presenting values 9% and 30% greater for Df and Dc DREff (Córdoba-Jabonero et al., 2021) and until 3 times greater for DD DREff (Granados-Muñoz et al., 2019); or even 15% lower for DD particles (Sicard et al., 2016).

By contrast with the DRE at BOA and TOA, the DRE is positive for the entire episode, indicating an atmospheric dust-induced heating, as DRE(BOA) > DRE(TOA) in absolute values (see Eq. 2). Regarding the episode average DRE, highest values are
found at ARN and GRA decreasing with latitude, i.e., 50% at EVO, 25% at TRJ and 65% at BCN (see Table 6). Moreover, the maximum hourly-averaged DRE value is found at ARN, being 55-80% lower for the rest of the stations (Table 6). Similar results are found in terms of the atmospheric ftr_DRE ratio, showing approximately values in the 27-33% range.

**4.3 Differences between approaches for DRE estimation in SW radiative flux simulations**

**4.3.1 Potential limitations in the radiative properties**

As described in Section 2.3, a comparative analysis is performed for the dust-induced DRE as obtained using two different approaches. That is, either the separated contribution of each of the two dust modes (DRE$^{(I)}$) or the contribution of the total dust overall (DRE$^{(II)}$) is considered. Note that the second approach is assumed as the reference in the comparison (see Eqs. 6 and 7) as widely used by other authors. Potential limitations in this methodology should be considered, as they could affect the differences in DRE simulations found between both approaches used in this study.

First, it should be taken into account that particular sensitivities to coarse and super coarse dust particles could be differentiated between passive sensors (as sun/sky photometers) and active ones (as lidar systems). Specifically, the properties of particles with radii exceeding 15 μm are not retrieved by the AERONET inversion algorithm. However, Ryder et al. (2019) demonstrated that dust particles with a radius greater than 15 μm contribute only 1-3% to the particle extinction coefficient at 550 nm. This indicates that this size cutoff effect in the AERONET data inversion procedure has a negligible impact on the

AERONET inversion products. Thus, the coarse components as regarded by AERONET and lidar inversion is comparable despite the potential different sensitivities between both sensors.

It is worth remembering another important difference by applying both approaches in the DRE simulations. In DRE$^{(I)}$ both $g$ and $\alpha$ are introduced for the fine and coarse modes separately, whereas in DRE$^{(II)}$ these variables are introduced for the total dust. Oppositely, AERONET $\omega$ is provided only for the total mode (no separation; see Table 1). To assess the impact of using

a single total $\omega$ instead of separate components, the parametrizations proposed by Wang et al., (2019) were applied. In that study, an efficient approach of the parameterization of dust aerosols in SW bands is provided. The differences between coarse $\omega$ obtained using the parametrizations and total $\omega$ of AERONET are less than 1% on average, except for the 440 nm wavelength, which rose to 3%. Those results are aligned with AERONET own uncertainty of $\omega$. Thus, using total $\omega$ in both Df and Dc DRE$^{(I)}$ simulations should not have a significant impact.

As far as the fine mode is concerned, the AERONET fine $g$ value is introduced in GAME for computing the DRE related to the Df component, as exposed in Section 2.1.1. However, it should be taken into account that the fine $g$ is influenced by the total fine mode, i.e. both Df and background aerosols. In addition, assuming a vertically uniform $g$ for the Df component could have substantial consequences, mainly because fine $g$ values can be strongly affected by background aerosols, which are also mostly confined to the boundary layer. Therefore, a complementary study has been conducted to study the degree of suitability

of applying AERONET fine $g$ values for DRE computation of the Df component. This way, hourly fine $g$ values for those cases reported under dust-dominated conditions (i.e., cases where the AERONET Fine Mode Fraction is less than 40%) were compared with respect to the fine $g$ values reported for all cases during the study period (25 March to 7 April 2021). In summary, results indicated that differences in fine $g$ between cases under dust-dominated and all conditions were not significantly high for those obtained at each station during the study period as well as regarding the period-averaged fine $g$

differences between stations. Specifically, the relative differences ranged between -3.4% and +0.4%. Therefore, it can be

assumed that the contribution of background aerosols to the fine $g$ can be considered negligible, and then its values can be used for the Df component. This outcome could be, to some extent, expected since background aerosols generally exhibit very low linear depolarization ratios, as they are predominantly small and spherical particles with a minimal contribution to the $g$ parameter. Hence, at least in this specific dust event, the assumption of a constant $g$ value for the Df component (i.e., AERONET fine $g$) throughout the entire atmospheric column can be considered reliable.

### 4.3.2 Comparative analysis

The comparison of the DRE computation at both BOA and TOA as obtained from the two approaches is based on examining their absolute ($\Delta DRE$; W m$^{-2}$) (see Eq. 6) and relative differences ($\Delta^{rel}DRE$; %) (see Eq. 7). On the one hand, Figure 7 shows $\Delta^{rel}DRE$ as a function of SZA from 25 March to 7 April 2021 for all five lidar stations involved in this study. The DD $DOD^{532}$ is also introduced, although a clear dependence is not observed. First, it should be noted that large $\Delta^{rel}DRE$ values are observed between methodologies at high SZA (> 70º), which is expected, as the intrinsic uncertainty in GAME simulations increases at these angles due to the assumption of a plane-parallel atmosphere in the model (see Sect. 2.1). Therefore, data at these SZAs (> 70º) should be discarded. Indeed, data percentiles $P(75) = +0.65\%$ and $P(25) = -0.85\%$ are obtained at both BOA and TOA, that is, only 25% out of $\Delta^{rel}DRE$ present values below -0.85% by considering all SZAs. However, those percentiles decreased at SZA < 70º, i.e., $P(75) = +0.02\%$ (-0.15%) and $P(25) = -6.8\%$ (-10.9%) are computed at BOA (TOA) (see Fig. 7). These results would indicate that lower absolute DRE$^{(I)}$ values with respect to DRE$^{(II)}$ (note that SW DRE is negative) are predominantly found. Therefore, dismissing the comparison for SZA > 70º, DRE$^{(I)}$ values on average are lower (in absolute units) with respect to DRE$^{(II)}$ (i.e., DRE$^{(I)}$ shows less negative values than DRE$^{(II)}$), with mean $\Delta^{rel}DRE$ around -3±5% and -6±10% on average at BOA and TOA, respectively. Therefore, DRE estimations that consider the separation of both dust Df and Dc components with respect to total dust would result in a less pronounced dust-induced cooling effect.

On the other hand, Figure 8 shows the absolute differences in DRE as obtained between the two approaches ($\Delta DRE$, see Eq. 6) as a function of DRE$^{(II)}$ at both BOA and TOA for all five lidar stations. The dependence on the DD $DOD^{532}$ is also highlighted. For SZA < 70º, approximately 75% out of $\Delta DRE$ values are positive, indicating that absolute DRE$^{(I)}$ values are lower than DRE$^{(II)}$ ones at both BOA and TOA. Indeed, $P(25) = -0.01$ (+0.03) W m$^{-2}$ at BOA (TOA) (see Fig. 8) is computed. Hence, those results indicate, overall, the DRE estimation by considering the separation of both dust components would result in a less pronounced cooling effect than that for the total dust (note the SW dust DRE is negative, i.e., dust induces a cooling effect at both BOA and TOA), as also stated previously. However, it should be noted that absolute $\Delta DRE$ tend to increase as $DOD^{532}$ increased, in general; $\Delta DRE$ were mostly close to zero at lower DOD (< 0.2), rather increasing at moderate/high dust load conditions (DOD > 0.50)) and reaching maximum values of nearly -20 W m$^{-2}$ at both BOA and TOA. In particular, when DRE$^{(II)}$ > -40 W m$^{-2}$ (point shown by a grey dashed line in Fig. 8), mean $\Delta DRE$ values are +0.4±0.8 and +0.3±0.8 W m$^{-2}$ for both BOA and TOA, respectively, meanwhile for DRE$^{(II)} \leq$ -40 W m$^{-2}$, corresponding to higher dust load conditions, $\Delta DRE$ show mean values of -2.8±4.0 and -3.7±4.3 W m$^{-2}$ at both BOA and TOA, respectively. Indeed, this is also reflected by

examining the significant quartiles (median) computed for $\Delta DRE$ when $DRE^{(II)} < -40$ W m$^{-2}$: $P(75)$ $(P(50)) = 0.17$ (-0.1) and -1.7 (-3.3) W m$^{-2}$ at BOA and TOA, respectively. These opposite results, depending on dust loading, suggest that dust induces a more pronounced cooling effect at both BOA and TOA under moderate to high dusty conditions when the contribution of the Dc and Df components are considered separately. Furthermore, this cooling effect is more evident at TOA than at BOA, probably due to the presence of dust at higher altitudes than those typically found in the troposphere.

The differences found between $DRE^{(I)}$ and $DRE^{(II)}$ can be explained by the uncertainties inherent to the radiative properties ($g$, $\omega$) introduced as inputs in the model. Andrews et al. (2006) determined with a sensitivity analysis that a 10% decrease in total $g$ would correspond to 13 and 19% reductions in BOA and TOA forcing, respectively. On the one hand, AERONET total $g$ and $\omega$ are given with an uncertainty of ±0.03 and ±0.04 (approximately 4% for both $g$ and $\omega$), respectively, for dust particles (Dubovik et al., 2000, 2002). On the other hand, it is worth examining the differences between the total and coarse $g$, considering that the contribution of Dc plays a significant role in DRE in cases with medium and high DOD. In particular, the coarse $g$ used in Dc simulations are, on average, from 8% to 10% greater than total $g$. Thus, the results found in our study agree with those obtained by Andrews et al. (2006).

Moreover, the Df role cannot be disregarded in DRE estimations. In this work, the impact of Df particles to DD DRE has been highlighted, even in cases with medium and high DOD, when the Dc contribution is plausibly dominating. It is possible that the Df contribution to the total DRE could be systematically underestimated by applying the $DRE^{(II)}$ approach in relatively high dusty environments.

## 5. Summary and conclusions

This work serves as a complement to the study conducted by López-Cayuela et al. (2023), where the vertical distribution of both the fine- and coarse-mode dust during an intense dust episode that took place between 25 March and 7 April 2021 crossing over the Iberian Peninsula was described in detail. Now, this study focuses on the estimation of temporal variations in the instantaneous and daily direct radiative effects of dust in the short-wave spectral range during the same dust event. The investigation is carried out using data obtained from five lidar stations situated in El Arenosillo/Huelva, Granada, Torrejón/Madrid and Barcelona in Spain, as well as Évora in Portugal, whose observations were also used in that previous study. The main findings are summarised next.

- Across all the stations examined in this study, the DRE remains negative throughout the dusty period at both BOA and TOA, indicating aerosol-induced cooling at these levels. Conversely, the DRE shows a positive trend, indicating an atmospheric aerosol-induced heating. The results are similar with those reported by a wide range of studies that can be found in the literature.

- On some specific days, DRE values were considerably high, corresponding to DODs greater than 0.80. Particularly, this work shows maximum hourly averaged DRE values of -54.1 W m$^{-2}$ (-96.3 W m$^{-2}$) and -35.9 W m$^{-2}$ (-53.5 W m$^{-2}$) as induced by Df (Dc) particles at BOA and TOA, respectively. Regarding the Df impact, they induce almost half

of the total DRE. Particularly, Df contributions of 40% and 45% are found at both BOA and TOA, respectively. Thus, despite dust intrusions are primarily governed by the coarse dust mode, the contribution of the fine mode plays a relevant role in DRE estimations.

- In terms of DREff, similar values are found at every station, with variations lesser than 10% between them. By comparing with other authors, this study shows DREff values slightly higher for Df particles (< 10%) at both BOA and TOA. However, the difference increases for Dc particles, with DREff values being a 15% and 30% greater at BOA and TOA, respectively, than those reported by other works. In terms of DD particles, the DREff results are similar to those reported at BOA, and higher than those at TOA.

- DRE simulations based on the separation of dust modes (after disregarding values for SZA > 70º) show that nearly 75% out of absolute DRE values are lower with respect to those obtained by considering the total dust (denoted as DRE$^{(II)}$ in this work). That is, the differences between the two approaches are mostly positive. This indicates that a less pronounced cooling effect would be induced overall from the DRE estimations based on the separation of the two dust components. However, the mean relative differences between approaches are not significant, i.e. -3% and -6% on average for SZA < 70º at both BOA and TOA, respectively.

- However, opposite results are obtained under moderate to high dust load conditions, with differences between approaches, as described in this work, mostly showing negative values. By separating the two dust components, DRE estimations suggest that dust induces a more pronounced cooling effect. This may highlight the important role of Df particles in DRE simulations, which are likely underestimated when total dust (no separation) is considered in relatively high dusty environments. In this study, that effect is more evident at the TOA than at the BOA, probably due to the presence of dust at higher levels than those usual in the troposphere. This fact can introduce relevant implications for radiometric measurements from satellite instrumentation.

For future work, it would be relevant to perform the same study, but under dusty conditions with rather variable ftr_DOD, to comprehensively assess the role of the fine dust particles in DRE by using two different conceptual approaches. This would allow determining how dust aging affects the DRE, as considered looking at the proportion of Df particles relative to DD. It is important to note that the methodology used in this work combines lidar data, which provide vertical aerosol profiles, with photometer data, which derive columnar aerosol radiative properties. One way to improve this methodology would be to use height-resolved key radiative parameters such as $\omega$ and $g$ profiles, which would be particularly useful in cases where either background aerosols play a significant role, or in scenarios involving a mixture of aerosol types. The use of multiwavelength lidars instead of single-wavelength lidar systems together with particular inversion methods could provide those required $\omega$ and $g$ profiles.

## Data availability

EARLINET lidar files are available from the EARLINET Data Portal (https://data.earlinet.org/, last access: 21 December 2021; Pappalardo et al., 2014). The accessibility of these files is limited based on the EARLINET criteria. Part of the data used in this publication was obtained as part of the AERONET network and is publicly available. For additional lidar data or information, please contact the corresponding author.

## Author contributions

MÁLC, CCJ and JLGR conceptualized the study. MÁLC, CCJ, MS and JLGR were responsible for the formal analysis. MÁLC wrote the original draft of the paper and applied the software. MÁLC, CCJ, MS, and JLGR carried out the investigation. MÁLC, CCJ, MS, VS, MJGM, AC, FTC, JABA, CMP, MJC, ARG, DB, JAG, LAA and JLGR reviewed and edited the paper. CCJ, MJGM, ARG and DB were responsible for data curation. CCJ, LAA, AC and MJC provided the resources. CCJ and JLGR supervised the investigation. All authors have read and agreed upon the published version of the paper.

## Competing interests

The contact author has declared that none of the authors has any competing interests.

## Acknowledgements

This work was funded by MCIN/AEI/10.13039/501100011033 and FEDER 'Una manera de hacer Europa' (grants PID2023-151666NB-I00, PID2020-117825GB-C21, PID2020-117825GB-C22, PID2020-120015RB-I00, PID2019-104205GB-C21, PID2019-103886RB-I00, EQC2018-004686-P, RED2022-134824-E), and supported by the University of Granada (grant A-RNM-430-UGR20, Singular Laboratory programme LS2022-1, Scientific Units of Excellence Program grant UCE-PP2017-02), and partially by EU H2020 (ACTRIS GA 871115) and MSC RISE (grant 101131631). PT team is co-funded by national funds through FCT in the framework of the ICT projects (grants UIDB/04683/2020 and UIDP/04683/2020). MS acknowledges the support of the European Commission through the REALISTIC project (GA 101086690). MÁLC is supported by the INTA predoctoral contract programme. MÁLC thanks to ATMO-ACCESS for the TNA LIRTASOM ('Lidar data in Radiative Transfer model for dust direct radiative effect estimation and evaluation against solar measurement') project, supported by the European Commission (H2020-INFRAIA-2020-1, grant 101008004). JAG thanks the Spanish Ministry of Universities for the grant FPU 21/01436. The BCN team thanks Ellsworth J. Welton for providing the MPL unit in place at the Barcelona site. Ellsworth J. Welton and Sebastian A. Stewart are warmly acknowledged for their continuous help in keeping the MPL systems up to date. Authors gratefully acknowledge the PIs and technical staff of all the lidar and AERONET stations for maintenance support of the instrumentation involved in this work.

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

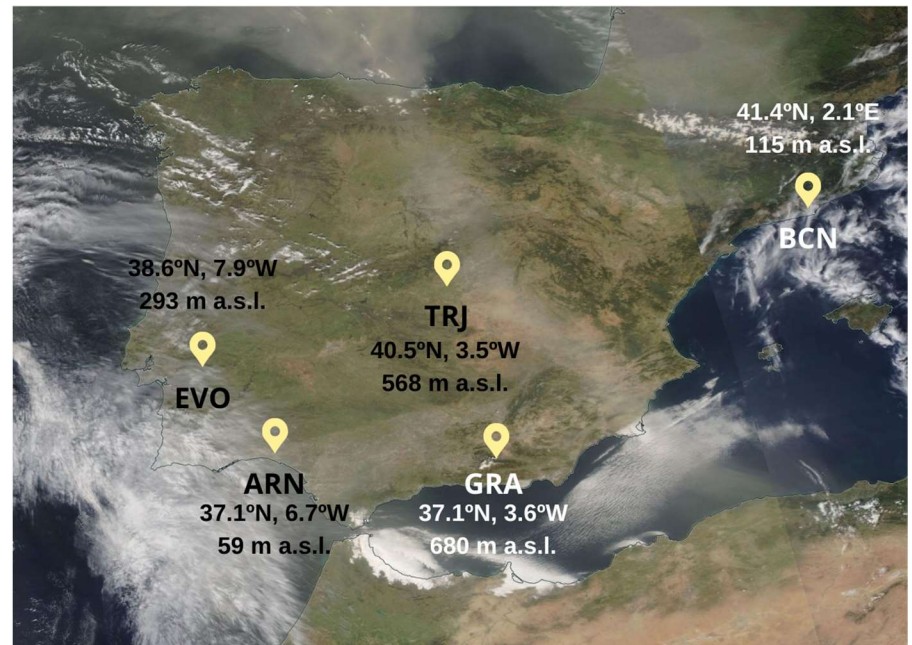

**Figure 1: MODIS Terra/Aqua corrected reflectance over the Iberian Peninsula on 31 March 2021. The five lidar stations are marked with a yellow pin. From south-west to north-east: El Arenosillo/Huelva (ARN), Évora (EVO), Granada (GRA), Torrejón/Madrid (TRJ) and Barcelona (BCN) sites. The coordinates and altitude are also included.**

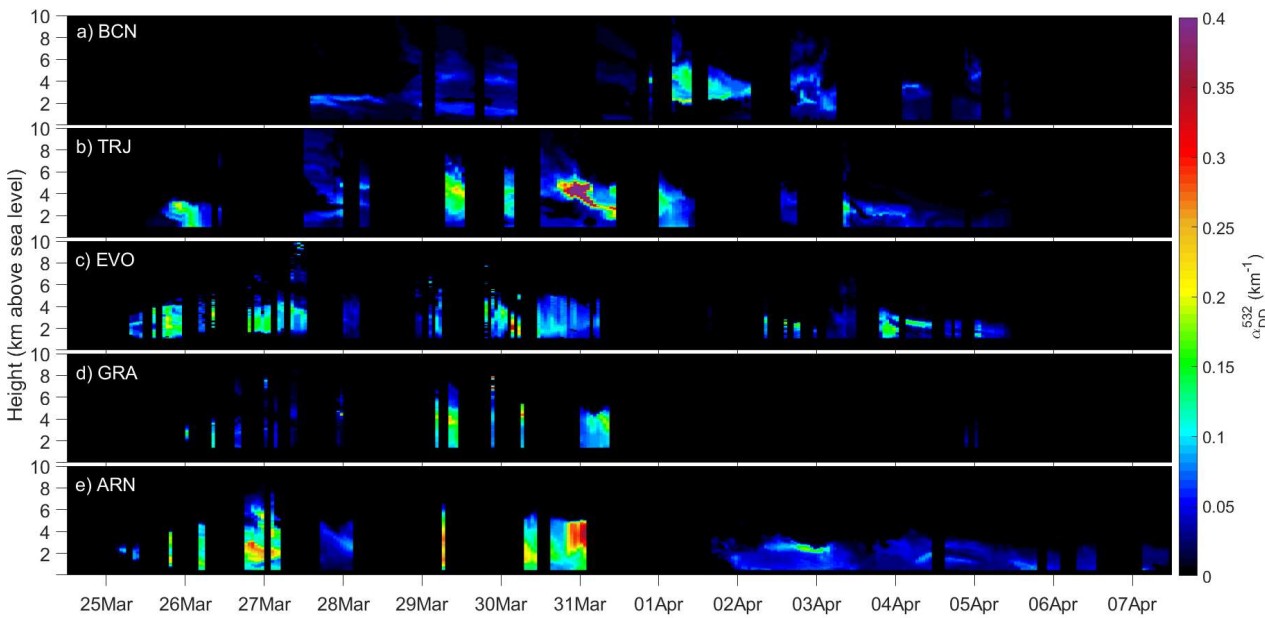

**Figure 2: Temporal evolution of the DD extinction coefficient ($\alpha_{DD}^{532}$, km$^{-1}$) at the five Iberian lidar stations (from North-East to South-West IP, by decreasing latitude): a) Barcelona (BCN), b) Torrejón/Madrid (TRJ), c) Évora (EVO), d) Granada (GRA) and e) El Arenosillo/Huelva (ARN). Profile gaps correspond to either no inversion available or no lidar measurements.**


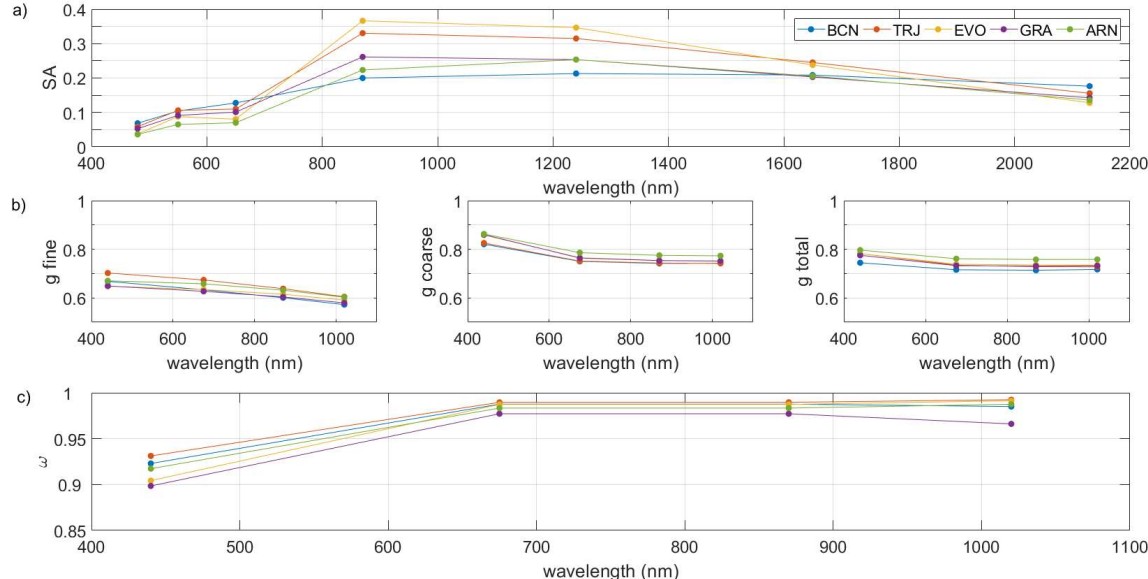

**Figure 3: Spectral behaviour of (a) the mean of the MODIS surface albedo (SA) distribution, (b) AERONET asymmetry factor (*g*) for fine, coarse and total modes, and (c) AERONET single scattering albedo (*ω*) at the five Iberian lidar stations: Barcelona (BCN, blue), Torrejón/Madrid (TRJ, orange), Évora (EVO, yellow), Granada (GRA, purple) and El Arenosillo/Huelva (ARN, green). The date and time chosen is 1 April at 10 UTC for BCN station and 31 March at 12 UTC for the rest of the sites.**


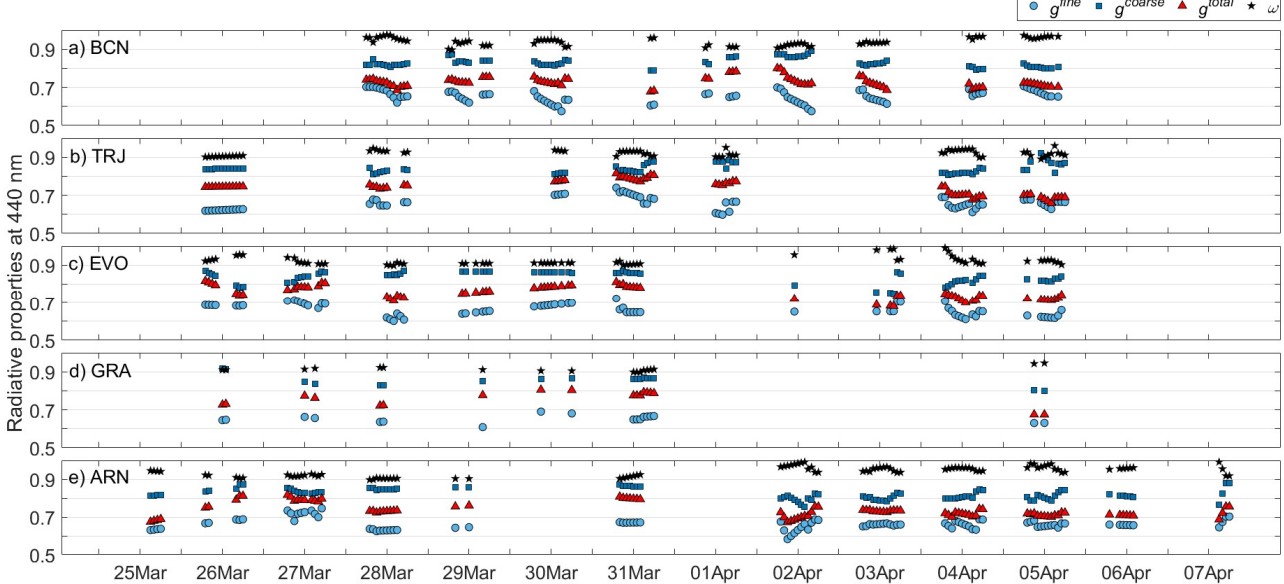

**Figure 4: Overview of the temporal evolution of the radiative properties of fine, coarse and total g, and for $\omega$ at 440 nm at the five lidar stations, i.e., (a) BCN, (b) TRJ, (c) EVO, (d) GRA and (e) ARN.**


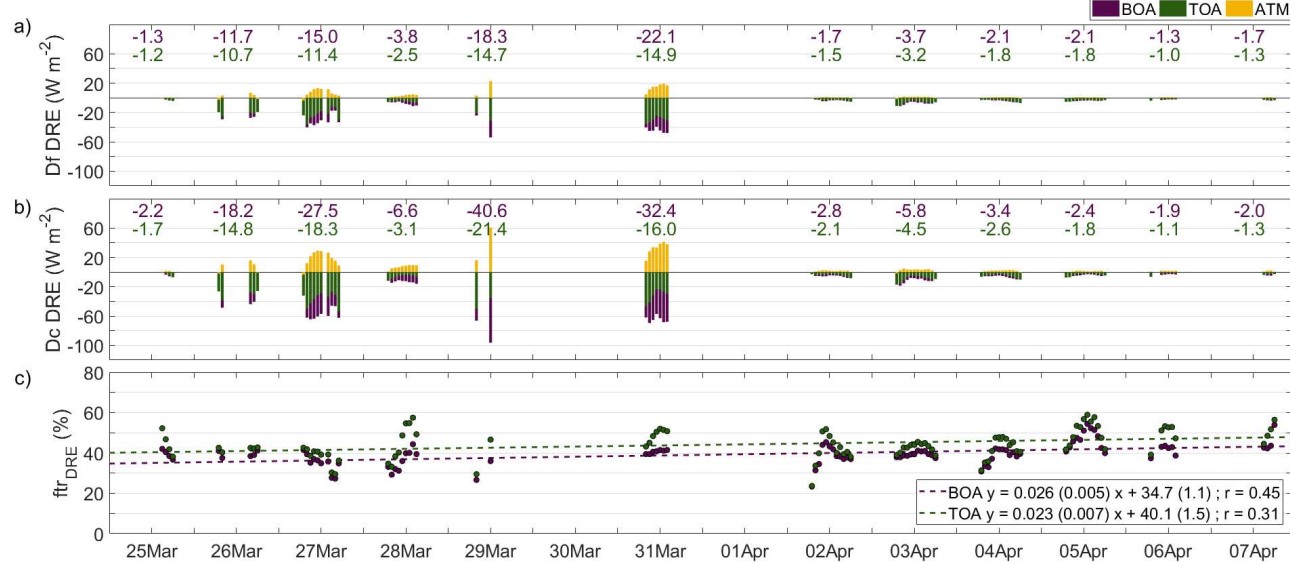

**Figure 5: Dust direct radiative effect (DRE, W m⁻²) at BOA (purple), TOA (green) and in the atmosphere (ATM, yellow) at the ARN station, for instance, as induced by the (a) fine dust (Df) and (b) coarse dust particles. Daily mean values are also included above (keeping the same colours). (c) Fine-to-total (Df/DD) ratio of the dust DRE (ftr_DRE) at BOA (purple) and TOA (green), together with their linear fit between 25 March and 7 April. The standard deviation of the slope and intercept are shown in brackets. The absolute temporal rate of the ftr_DRE ratio (δDRE) at BOA (TOA) is +0.026% h⁻¹ (+0.023 % h⁻¹), being equivalent to an increase of +0.62 % day⁻¹ (+0.55 % day⁻¹).**

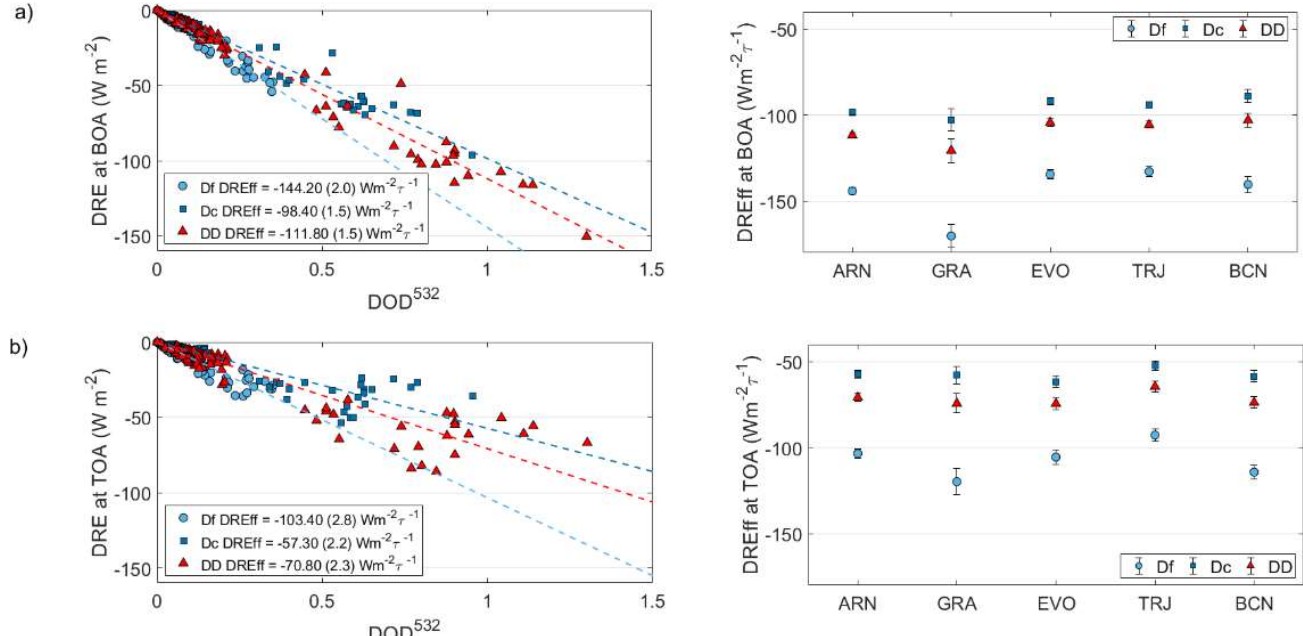

**Figure 6: Dust direct radiative effect (DRE) as a function of DOD[532], for instance, at ARN station (left panels), together with the DREff values as obtained for each station (right panels), at a) BOA, and b) TOA. Both are shown separately for the fine dust (Df, blue circles), coarse dust (Dc, dark blue squares) and total dust (DD, red triangles). The error of the DREff are shown in brackets in the left panels, and by bars in the right panels. The period consider is from 25 March to 7 April 2021.**

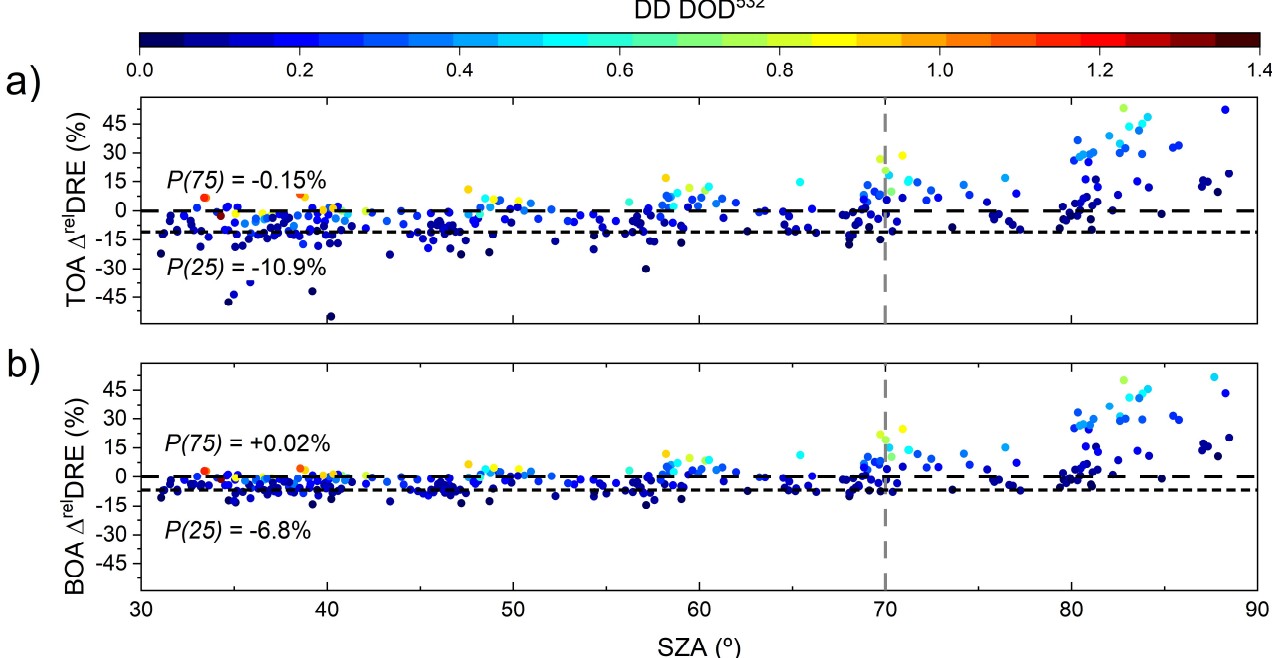

**Figure 7. Relative differences in DRE ($\Delta^{rel}DRE$; %) as obtained between the two approaches as a function of SZA at: a) BOA and b) TOA for all the five lidar stations involved in this study, from 25 March to 7 April 2021. The dependence on DD $DOD^{532}$ is shown by a colour-scaled bar. $P(25)$ and $P(75)$ are marked by a horizontal short-dashed and dashed line, respectively. The vertical grey dashed line denotes SZA = 70°.**

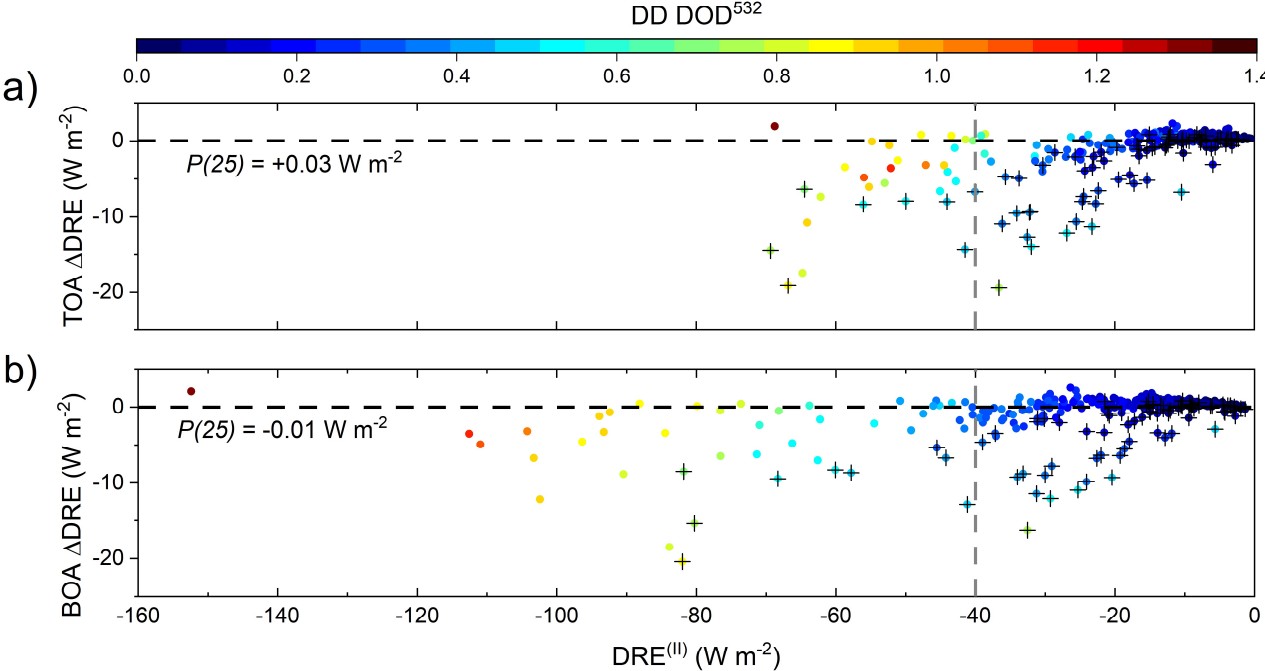

Figure 8. Differences in DRE ($\Delta DRE$; W m⁻²) as obtained between the two approaches (DRE[I] - DRE[II]; see Eq. 6) at: a) BOA, and b) TOA for all the five lidar stations involved in this study, from 25 March to 7 April 2021. The dependence on the DD $DOD^{532}$ is shown by a colour-scaled bar. $P(25)$ is marked by a horizontal dashed line. Data for SZA > 70° are highlighted with a cross symbol. Vertical grey dashed line denotes the point at DRE[II] = -40 W m⁻².

**Table 1: Input parameters and radiative properties for the GAME model and databases in the SW spectral range. Note that $z$ denotes the vertical dependence. DRE$^{(I)}$ and DRE$^{(II)}$ denotes the particular approach used for DRE simulation, considering either dust component separation (Dc, Df) or total dust (DD, no separation), respectively.**

| | Parameters | | Database / instrumentation |
|---|---|---|---|
| | DRE$^{(I)}$ | DRE$^{(II)}$ | |
| Atmosphere and Land | SA | | MODIS |
| | Atmospheric profiles | | U.S. std. atm. + 3h GDAS profiles |
| | Gas concentration profiles | | U.S. std. atm. + 3h GDAS profiles |
| | Absorption coefficients | | HITRAN |
| Aerosols | Df or Dc $\alpha^{532}(z)$ | DD $\alpha^{532}(z)$ | Lidar |
| | Df or Dc DOD | DD DOD | Lidar |
| | fine or coarse $g$ | total $g$ | Sun-photometer |
| | total $\omega$ | | Sun-photometer |
| | $AE^{440-870}$ | | Sun-photometer |

800

**Table 2: Dust optical depth at 532 nm (DOD$^{532}$) for the dust fine (Df) and coarse (Dc) components and DD (DD = Df + Dc) at the five lidar stations (from left to right as latitude increases). ftr_DOD denotes the Df-to-DD ratio (in %). $\overline{X}$ indicates the average for the whole dust event (the standard deviation is also shown) and $X^{max}$ refers to the maximal hourly value within the dusty period.**

|  | ARN | GRA | EVO | TRJ | BCN |
|---|---|---|---|---|---|
| $\overline{Df}$ | $0.10 \pm 0.10$ | $0.08 \pm 0.06$ | $0.06 \pm 0.03$ | $0.08 \pm 0.07$ | $0.04 \pm 0.02$ |
| $\overline{Dc}$ | $0.24 \pm 0.25$ | $0.20 \pm 0.16$ | $0.15 \pm 0.08$ | $0.19 \pm 0.15$ | $0.10 \pm 0.06$ |
| $\overline{DD}$ | $0.34 \pm 0.35$ | $0.28 \pm 0.22$ | $0.21 \pm 0.11$ | $0.27 \pm 0.22$ | $0.14 \pm 0.08$ |
| $ftr\_DOD$ (%) | 29.4 | 28.6 | 30.0 | 28.5 | 28.6 |
| $Df^{max}$ | 0.35 | 0.18 | 0.19 | 0.27 | 0.16 |
| $Dc^{max}$ | 0.95 | 0.49 | 0.42 | 0.61 | 0.38 |
| $DD^{max}$ | 1.30 | 0.65 | 0.61 | 0.88 | 0.54 |

**Table 3: Radiative properties at 440 nm at the five lidar stations (from left to right as latitude increases). Particularly, the parameters are: the dust fine (Df) and coarse (Dc) components of the asymmetry factor ($g$); and the total dust component for both $g$ and single scattering albedo ($\omega$). The symbol $\overline{X}$ indicates the average for the whole dust event (the standard deviation is also shown) and $X^{max}$ and $X^{min}$ refers to the maximal and minimal hourly value within the dusty period, respectively.**

| | | ARN | GRA | EVO | TRJ | BCN |
|---|---|---|---|---|---|---|
| $g^{440}$ | $\overline{Df}$ | 0.662 ± 0.029 | 0.652 ± 0.020 | 0.661 ± 0.031 | 0.658 ± 0.033 | 0.652 ± 0.032 |
| | $Df^{max}$ | 0.746 | 0.691 | 0.722 | 0.740 | 0.706 |
| | $Df^{min}$ | 0.575 | 0.599 | 0.601 | 0.599 | 0.575 |
| | $\overline{Dc}$ | 0.822 ± 0.028 | 0.856 ± 0.030 | 0.837 ± 0.031 | 0.841 ± 0.025 | 0.829 ± 0.024 |
| | $Dc^{max}$ | 0.881 | 0.917 | 0.870 | 0.920 | 0.892 |
| | $Df^{min}$ | 0.755 | 0.798 | 0.750 | 0.807 | 0.788 |
| | $\overline{total\ dust}$ | 0.735 ± 0.037 | 0.759 ± 0.041 | 0.754 ± 0.035 | 0.740 ± 0.038 | 0.728 ± 0.026 |
| | $total\ dust^{max}$ | 0.817 | 0.805 | 0.816 | 0.815 | 0.800 |
| | $total\ dust^{min}$ | 0.672 | 0.675 | 0.681 | 0.659 | 0.678 |
| $\omega^{440}$ | $\overline{total\ dust}$ | 0.941 ± 0.026 | 0.914 ± 0.013 | 0.922 ± 0.022 | 0.921 ± 0.016 | 0.941 ± 0.021 |
| | $total\ dust^{max}$ | 0.992 | 0.946 | 0.990 | 0.960 | 0.976 |
| | $total\ dust^{min}$ | 0.897 | 0.898 | 0.897 | 0.890 | 0.897 |

**Table 4: Episode-averaged hourly values of the dust radiative effect (DRE, in W m⁻²) and their standard deviation on the surface (BOA) as induced by fine (Dc), coarse (Dc) and total dust (DD) at the five Iberian lidar stations. $\overline{X}$ indicates the average for the whole event (standard deviations are also shown), and $X^{max}$ refers to the maximal value. The DREff (in W m⁻² τ⁻¹) denotes the DRE efficiency. The ftr_DRE denotes the Df-to-total dust ratio (in %,) and $\delta DRE$ (in % day⁻¹) is the slope of the linear fitting analysis of the hourly averaged ftr_DRE along time, considering a minimum of six ftr_DRE each day.**

|  |  | ARN | GRA | EVO | TRJ | BCN |
|---|---|---|---|---|---|---|
| Df | $\overline{DRE}$ | -7.1 ± 7.6 | -5.9 ± 4.6 | -3.8 ± 1.9 | -5.9 ± 3.8 | -3.1 ± 1.9 |
|  | $DRE^{max}$ | -54.1 | -28.8 | -17.8 | -32.5 | -27.9 |
|  | DREff | -144.2 ± 2.0 | -170.3 ± 6.5 | -134.5 ± 2.6 | -132.9 ± 2.8 | -140.4 ± 4.8 |
| Dc | $\overline{DRE}$ | -12.2 ± 13.9 | -10.4 ± 8.6 | -6.3 ± 3.5 | -9.6 ± 6.6 | -5.2 ± 3.1 |
|  | $DRE^{max}$ | -96.3 | -52.2 | -28.0 | -55.4 | -41.7 |
|  | DREff | -98.4 ± 1.5 | -102.9 ± 6.6 | -92.0 ± 2.0 | -94.0 ± 1.7 | -89.0 ± 3.7 |
| DD | $\overline{DRE}$ | -19.2 ± 21.4 | -16.0 ± 13.2 | -10.1 ± 5.4 | -15.5 ± 10.3 | -8.3 ± 5.0 |
|  | $DRE^{max}$ | -150.3 | -77.6 | -45.9 | -87.9 | -69.6 |
|  | DREff | -111.8 ± 1.5 | -120.7 ± 6.8 | -104.4 ± 2.1 | -105.7 ± 2.0 | -103.0 ± 4.0 |
| frt_DRE |  | 38.6 | 38.0 | 41.5 | 38.5 | 37.6 |
| $\delta DRE$ |  | 0.62 | 0.02 | 0.43 | 0.77 | 0.26 |

**Table 5: The same as Table 4, but on the top-of-the-atmosphere (TOA).**

|  |  | ARN | GRA | EVO | TRJ | BCN |
|---|---|---|---|---|---|---|
| Df | $\overline{DRE}$ | -5.5 ± 5.6 | -4.0 ± 3.7 | -2.8 ± 1.6 | -4.2 ± 2.8 | -2.5 ± 1.6 |
|  | $DRE^{max}$ | -35.9 | -28.8 | -17.7 | -23.4 | -23.2 |
|  | DREff | -103.4 ± 2.8 | -119.8 ± 7.6 | -105.5 ± 4.1 | -92.7 ± 3.6 | -114.2 ± 3.9 |
| Dc | $\overline{DRE}$ | -7.4 ± 7.8 | -5.2 ± 5.8 | -3.9 ± 2.8 | -5.3 ± 4.2 | -3.4 ± 2.2 |
|  | $DRE^{max}$ | -53.5 | -27.6 | -29.2 | -35.8 | -30.6 |
|  | DREff | -57.3 ± 2.2 | -57.9 ± 5.0 | -61.6 ± 3.5 | -52.4 ± 2.8 | -58.4 ± 3.2 |
| DD | $\overline{DRE}$ | -12.9 ± 13.3 | -9.2 ± 9.4 | -6.7 ± 4.3 | -9.5 ± 7.1 | - 6.0 ± 3.7 |
|  | $DRE^{max}$ | -85.9 | -48.1 | -47.1 | -58.6 | -51.7 |
|  | DREff | -70.8 ± 2.3 | -74.3 ± 5.6 | -74.4 ± 3.6 | -64.5 ± 3.0 | -73.8 ± 3.3 |
| frt_DRE |  | 43.8 | 45.5 | 42.2 | 44.6 | 43.3 |
| $\delta DRE$ |  | 0.55 | -2.90 | 0.67 | 2.64 | 0.70 |

830

**Table 6: The same as Table 4, but in the atmosphere (ATM).**

|  |  | ARN | GRA | EVO | TRJ | BCN |
|---|---|---|---|---|---|---|
| Df | $\overline{DRE}$ | $1.5 \pm 2.0$ | $1.9 \pm 1.2$ | $1.0 \pm 0.5$ | $1.5 \pm 1.2$ | $0.6 \pm 0.4$ |
|  | $DRE^{max}$ | 22.9 | 8.9 | 7.9 | 12.1 | 5.1 |
| Dc | $\overline{DRE}$ | $4.4 \pm 6.5$ | $4.9 \pm 3.1$ | $2.4 \pm 1.2$ | $3.7 \pm 2.9$ | $1.8 \pm 1.2$ |
|  | $DRE^{max}$ | 60.6 | 25.7 | 17.6 | 26.0 | 13.3 |
| DD | $\overline{DRE}$ | $5.9 \pm 8.4$ | $6.8 \pm 4.2$ | $3.4 \pm 1.7$ | $5.2 \pm 4.1$ | $2.4 \pm 1.6$ |
|  | $DRE^{max}$ | 83.5 | 33.0 | 25.6 | 38.1 | 18.0 |
| ftr_DRE |  | 27.4 | 28.5 | 28.8 | 29.6 | 25.0 |

835