# Peer review of "Fine and coarse dust radiative impact during an intense Saharan dust outbreak over the Iberian Peninsula. Short-wave direct radiative effect"

_EGUsphere, 2024_

## Author Comment (AC2)

The authors highly appreciate the thorough review provided by all the reviewers that has significantly improved the quality of the manuscript. For clarity, the revisions addressing the comments from Reviewers #2 (RC1), #3 (RC2), and #4 (RC3) are highlighted in blue, purple, and green, respectively, and included in one single file because of common comments. Changes that respond to comments from multiple reviewers are marked in red. In addition, Table 1 has been slightly modified, Figure 2 has been improved, and the previous Figure 7 has been replaced by the new two Figures 7 and 8 in order to improve the reading and present more clearly the main results (Table 1 and all these Figures are included at the end). The Abstract has also been modified accordingly with the changes introduced in the manuscript (lines 27-34), and the rest of manuscript has been substantially revised by accounting for all the reviewers' suggestions, comments and recommendations.
* * *
**Reviewer #2's comments**

This study quantifies the dust direct radiative effect (DRE) in the short-wave range (SW) during a springtime dust episode over the Iberian Peninsula using data from five lidar stations. The authors emphasize the comparison of two distinct methodologies for estimating the SW DRE: one that directly calculates the DRE of dust, and another that calculates the DRE separately for fine and coarse dust, which are then summed to provide the total DRE of dust. The study highlights that the fine fraction, which primarily modulates the SW DRE, cannot be disregarded, contributing about 50% of the total DRE at the top of the atmosphere (TOA) and bottom of the atmosphere (BOA). The differences between the two approaches are linked to variations in the assumed asymmetry factor (derived from AERONET inversions) between the fine mode and total dust.

General Comments:

What sets this study apart from previous research is its comparison of the two different methodologies. In previous studies, the authors have used the methodology that separates the fine and coarse components, and in this study, they compare this approach with the more commonly used method based on lidar measurements to estimate the DRE. While the study is interesting—such separation can be very useful for evaluating DRE per size mode in dust climate models—I have several major comments that need to be addressed before considering the paper for publication.

The main novelty of the paper, as highlighted in the abstract, is the comparison of the two methodologies. However, the paper is very descriptive and fails to provide a comprehensive assessment of the causes behind the discrepancies between the two methods. Most of the paper (section 4.2 and associated figures) is devoted to describing the episode in terms of the evolution of the dust properties and DRE across the stations considered and comparing these findings with previous studies and other events in the region. Only a very short section (4.3) is dedicated to exploring the differences between the methods. While the differences are highlighted, they are not explored in detail, leaving the conclusions and implications of these results unclear.

Several aspects need to be considered in the analysis of the results to provide a comprehensive picture:

1. A key difference between both methods is the assumed asymmetry factor in the fine, coarse and total dust derived directly from AERONET inversions. These asymmetry factors are supposed to be internally consistent with other AERONET inversion products (PSD, fine and coarse mode AOD and SSA). In other words, one should be able to derive the asymmetry factor of the total dust derived from AERONET from the asymmetry factors of the fine and coarse components weighted by the AOD and the SSA of the fine and coarse components, respectively. If that is true, it may not be surprising that the DRE calculated using the asymmetry factors of the fine and coarse components together with the fine and coarse dust extinction from the lidar measurements differs from the DRE calculated directly from

the asymmetry factor of the total dust and the overall extinction from the lidar measurements. Understandably, this difference seems to scale with the differences between the asymmetry factor of the fine mode and that of the total dust. All this points towards the lack of consistency between AERONET inversions (which are internally consistent) and the lidar retrievals (fine and coarse extinction). This aspect needs to be explored in much more detail. First, the AOD of the fine and coarse components from the AERONET measurements and the ones from the lidar measurements should be compared. To what extent the differences in the fine to coarse ratio of the AOD between AERONET and the lidars can explain the results? One potential sensitivity test would be to constrain the fine and coarse dust extinction (and the total extinction) of the lidars with the AODs of AERONET and then calculate the associated DREs.

**Authors' Response:** Please, see the answer to the #9 and #13 specific comments, which are related to this concern.

2.  The above is just an example of the multiple analyses that could be done to comprehensively understand the discrepancies between the two methods. In addition, there are other aspects that are not properly discussed: 1) are the coarse components of AERONET and lidar comparable given the potential different sensitivities to coarse and super coarse dust particles between active and passive sensors? This is particularly important in this case given that the differences in the retrieved fine and coarse components of the extinction may at least partly explain the differences between the methods. 2) To what extent the asymmetry factors for the fine component in AERONET are affected by anthropogenic aerosol in the boundary layer? Given this influence, is it wise to assign these asymmetry factors to fine dust?

**Authors' Response:**

2.1 It is true that there are potential different sensitivities to coarse and super coarse dust particles between passive sensors (as sun/sky photometers) and active ones (as lidar systems). Specifically, the properties of any particle with radii exceeding 15 µm are not retrieved by the AERONET inversion algorithm. However, Ryder et al. (2019) demonstrated that dust particles with a radius greater than 15 µm contribute only 1%-3% to the particle extinction coefficient at 550 nm. This indicates that this size cutoff effect in the AERONET data inversion procedure has a negligible impact on the AERONET inversion products. Thus, the coarse component as regarded by AERONET and lidar inversion is comparable despite the potential different sensitivities between both sensors.

2.2 Please, see the answer in the #9 specific comment related to this concern.

In both cases, the text has been modified in the manuscript (page 9, lines 384-390).

Reference:
- Ryder, C. L., Highwood, E. J., Walser, A., Seibert, P., Philipp, A., and Weinzierl, B.: Coarse and giant particles are ubiquitous in Saharan dust export regions and are radiatively significant over the Sahara, Atmos. Chem. Phys., 19, 15353–15376, doi:10.5194/acp-19-15353-2019, 2019.

3.  All in all, my main suggestion is to reduce the more descriptive parts of the paper (section 4.2) and emphasize more on the analysis of the differences between the methods (through hypothesis testing) in section 4.3 along the lines highlighted above. More elaborated conclusions and implications should be considered.

**Authors' Response:** The text has been modified following the Reviewer's main suggestions. Please, see the answer to the specific comments for further details in Sections 4.2 (pages 10-12, lines 306-376) and 4.3 (pages 12-14, lines 377-446).

Specific comments:

4.  Line 29: How relevant is to mention this in the abstract? Who has ever said that the fine mode could be disregarded?

**Authors' Response:** We agree that the previous text was confusing. The original sentences: '*Results agree with the fact that the Df role cannot be disregarded, that is primarily responsible for SW radiative modulation. In particular, Df contributed nearly half of the total DRE at BOA and TOA in this event*' have been modified to: '*Results reveal that the Df role must be highlighted, as Df particles contributed nearly half of the total SW DRE at BOA and TOA, particularly for this event.*' (Abstract, lines 27-28).

5.  Line 45: Given the focus of the study better emphasize on the uncertainties in the SW (which are by the way quite high and very important given the strong SW forcing).

**Authors' Response:** To address this suggestion, the original sentence: "*Significant uncertainties remain in the estimation of the dust radiative effect, mostly because of the lack of observational constraints in dust interactions with clouds, among other factors (Kok et al., 2023).*", has been replaced by: "*Significant uncertainties remain in the estimation of the dust radiative effect, mostly because of the lack of observational constraints. Regarding the SW range, the balance between scattering and absorption is determined by the dust particle size and mineralogy, and the underlying surface's albedo determines the extent to which both processes impact the TOA radiative flux (Kok et al., 2023).*" (Page 2, lines 44-47).

6.  Line 50: Can you provide a reference for the relationship of heatwaves and dust intrusions?

**Authors' Response:** Regarding this suggestion, several references have been added, and also the following text has been included (page 2, lines 49-58):

'*On one hand, the geographical proximity of the Iberian Peninsula (IP) to North Africa, as well as the persistence of certain favourable weather patterns (Russo et al., 2020; Couto et al., 2021), make the IP one of the main pathway regions for Saharan desert dust transport towards Europe. On the other hand, Sánchez-Benítez et al. (2020) found that Iberian heatwave events are primarily linked to anomalous atmospheric circulation, typically characterized by a pronounced positive 500 hPa geopotential height anomaly (Z500) aloft. In particular, an analysis of daily weather regimes during Iberian heatwave days reveals a marked dominance of positive Z500 anomalies over western Europe, resembling the occurrence of Euro-Atlantic subtropical ridges. The frequency of this occurrence during heatwave days is observed to double relative to climatological averages. Last studies linked both processes, showing robust evidence for increases in maximum temperatures and the frequency of heatwaves over Europe (IPCC, 2023), which can be partially associated with dust intrusions (Sousa et al., 2019; Fernandes and Fragoso, 2021; Barriopedro et al., 2023).*'

References:

▪ Barriopedro, D., García-Herrera, R., Ordóñez, C., Miralles, D. G., and Salcedo-Sanz, S.: Heat waves: Physical understanding and scientific challenges, Rev. Geophys., 61, e2022RG000780, doi:10.1029/2022RG000780, 2023.

▪ Sánchez-Benítez, A., Barriopedro, D., García-Herrera, R.: Tracking Iberian heatwaves from a new perspective, Weather Clim. Extrem., 28, 100238, doi:j.wace.2019.100238, 2020.

▪ Sousa, P. M., Barriopedro, D., Ramos, A. M., García-Herrera, R., Espírito-Santo, F., and Trigo, R. M.: Saharan air intrusions as a relevant mechanism for Iberian heatwaves: The record breaking events of August 2018 and June 2019, Weather Clim. Extrem., 26, 100224, doi:10.1016/j.wace.2019.100224, 2019.

▪ Fernandes, R., and R. Fragoso, M.: Assessing Heatwaves and Their Association with North African Dust Intrusions in the Algarve (Portugal), Atmosphere, 12(9), 1090, doi:10.3390/atmos12091090, 2021.

7. Line 71: In which studies the effect of the fine dust fraction has been ignored? I think this is not true. You may mean that studies may have not separated the fine and coarse contributions. When using the total extinction, one is accounting for fine dust as well.

**Authors' Response:** We totally agree. We have rewritten the original sentence: '*in which the potential impact of Df particles on the total radiative effect has been mostly ignored*', which has been changed to: '*in which the potential impact of Df particles on the total radiative effect has not been separately underlined*' (page 3, lines 78-79).

8. Line 80 to 84: Can you explain the relevance of this in a broad context. Why is this important.

**Authors' Response:** It should be considered that although dust events predominantly carry coarse particles, the amount of fine dust particles could be screened within the total fine mode (e.g. fine dust + non-dusty aerosols). By using the methodology proposed in this work (see Sect. 2), we can separate three components: coarse dust, fine dust and non-dust, and then derive the DRE induced by each one separately. On one hand, the Df contribution could be also masked when the dust is considered as a whole (total dust), as the radiative properties, which are introduced in the radiative transfer model, could be slightly biased to those values corresponding to the coarse mode. This can be especially relevant for data assimilation in dust climate models. On the other hand, recent studies highlight the fact that Dc is inaccurately represented in global atmospheric models (Adebiyi and Kok, 2020; and references therein).

Therefore, in order to highlight these aspects, the following text has been added: '*It should be highlighted that in dust climate models, the separation can be particularly helpful for analysing DRE per size mode. On one hand, when dust is treated as a singular entity (total dust), the contribution of fine dust can be masked, as the radiative properties input into radiative transfer models tend to be skewed towards the characteristics of coarse dust. On the other hand, recent studies have found evidence that Dc is inadequately represented in global atmospheric models (Adebiyi and Kok, 2020). In particular, the mass of coarse dust in the atmosphere could be about four times greater than simulated in current AeroCom climate models (https://aerocom.met.no/). Since Dc warms by absorbing both SW and LW radiation (Kok et al., 2017), the underestimation of Dc by both climate models indicate that the net DRE could be more warming than has been previously estimated. Overall, the atmosphere appears to contain approximately 40% more dust (both Df and Dc) than what is simulated by AeroCom models, accounting for about 80% of the total particulate mass load in the atmosphere (Textor et al., 2007; Adebiyi and Kok, 2020). Thus, Df and Dc distinction is particularly relevant for data assimilation processes in climate models, where accurately representing the radiative properties of different dust components is critical for improving model performance and predictions.*' (Page 3, lines 88-99).

References added:
- Adebiyi, A. A., and Kok, J. F.: Climate models miss most of the coarse dust in the atmosphere, Science Adv., 6, 15, doi:10.1126/sciadv.aaz9507, 2020.
- Kok, J. F., Ridley, D. A., Zhou, Q., Miller, R. L., Zhao, C., Heald, C. L., Ward, D. S., Albani, S., Haustein, K.,: Smaller desert dust cooling effect estimated from analysis of dust size and abundance, Nat. Geosci., 10, 274–278, doi:10.1038/ngeo2912, 2017.
- Textor, C., Schulz, M., Guibert, S., Kinne, S., Balkanski, Y., Bauer, S., Berntsen, T., Berglen, T., Boucher, O., Chin, M., Dentener, F., Diehl, T., Feichter, J., Fillmore, D., Ginoux, P., Gong, S., Grini, A., Hendricks, J., Horowitz, L., Huang, P., Isaksen, I. S. A., Iversen, T., Kloster, S., Koch, D., Kirkevåg, A., Kristjansson, J. E., Krol, M., Lauer, A., Lamarque, J. F., Liu, X., Montanaro, V., Myhre, G., Penner, J. E., Pitari, G., Reddy, M. S., Seland, Ø., Stier, P., Takemura, T., Tie, X.: The effect of harmonized emissions on aerosol properties in global models – an AeroCom experiment, Atmos. Chem. Phys., 7, 4489–4501, doi:10.5194/acp-7-4489-2007, 2007.

9. Line 108: The assumption of vertically constant g in the fine mode could have profound implications given the effect of boundary layer aerosols. This should be highlighted here and in the discussion of the results.

**Authors' Response:** Introducing vertically constant $g$ values, as derived from the columnar AERONET products, into the model can have a notable impact, particularly for the fine mode, due to the additional presence of background (fine) aerosols at the boundary layer. However, in this specific event, the contribution of these background aerosols to the fine-mode $g$ values is considered negligible, as explained in the paragraph below. Therefore, this assumption does not significantly affect the results obtained.

An additional analysis was conducted to support such a statement, with the key findings presented in Section 4.3, which has been divided into two subsections for clarity. The text has been revised accordingly (page 13, lines 399-414):

'As far as the fine mode is concerned, the AERONET fine $g$ value is introduced in GAME for computing the DRE related to the Df component, as exposed in Section 2.1.1. However, it should be taken into account that the fine $g$ is influenced by the total fine mode, i.e. both Df and background aerosols. In addition, assuming a vertically uniform $g$ for the Df component could have substantial consequences, mainly because fine $g$ values can be strongly affected by background aerosols, which are also mostly confined to the boundary layer. Therefore, a complementary study has been conducted to study the degree of suitability of applying AERONET fine $g$ values for DRE computation of the Df component. This way, hourly fine $g$ values for those cases reported under dust-dominated conditions (i.e., cases where the AERONET Fine Mode Fraction is less than 40%) were compared with respect to the fine $g$ values reported for all cases during the study period (25 March to 7 April 2021). In summary, results indicated that differences in fine $g$ between cases under dust-dominated and all conditions were not significantly high for those obtained at each station during the study period as well as regarding the period-averaged fine $g$ differences between stations. Specifically, the relative differences ranged between -3.4% and +0.4%. Therefore, it can be assumed that the contribution of background aerosols to the fine g can be considered negligible, and then its values can be used for the Df component. This outcome could be, to some extent, expected since background aerosols generally exhibit very low linear depolarization ratios, as they are predominantly small and spherical particles with a minimal contribution to the $g$ parameter. Hence, at least in this specific dust event, the assumption of a constant $g$ value for the Df component (i.e., AERONET fine $g$) throughout the entire atmospheric column can be considered reliable.'

10. General aspects of section 2.1: a more comprehensive description of the AERONET products is needed. Please consider my general comments here on the internal consistency of the products, the potential limitations for the coarse mode, and the potential effect of anthropogenic aerosols in the fine mode. Also, better describe the assumptions in the lidar retrievals (even if they are provided in other publications) in comparison to AERONET. It is a good moment to talk about the potential inconsistencies when using fine and coarse g from AERONET together with the lidar extinctions.

**Authors' Response:** Section 2.1 has been modified and divided into two subsections (*2.1.1 AERONET properties used as input in GAME*, and *2.1.2 Lidar-derived extinction used as input in GAME*; pages 4-5, lines 122-152) taking into account the Reviewer #2's comments. A wider explanation about POLIPHON methodology and its uncertainties has been added. Moreover, the potential limitations in the radiative properties have been also explained in Section 4.3.1 (pages 12-13, lines 378-414).

11. Line 154: Why it could be considered more precise? I do not understand why. This cannot be shown. For example, the potential inconsistency between the fine and coarse modes in AERONET and the lidars may make the more refine method even more uncertain.

**Authors' Response:** Agree, 'precise' is a severe word not suitable to be used in this context. Then, the text has been changed as follows:

*'However, the novelty of this study lies in showing that the first approach (DRE^(I)) enables the dust DRE computation by using the optical properties of each dust mode separately (Sicard et al., 2014b; Córdoba-Jabonero et al., 2021; Sicard et al., 2022) assuming that the dust-induced diffuse radiation from one of the modes does not interact with the other one.'*, (page 7, lines 199-201).

12. Line 175: dust ageing is typically used for chemical ageing. Also, what do you mean by absences of uniform gravitational settling?

**Authors' Response:** We used 'dust ageing' as one of the mechanisms particularly linked to changes in the Df/DD proportion by gravitational settling along the dust pathway across the IP. We agree it must be confusing, so the sentence *'the absence of dust ageing observed throughout the IP'* has been changed to *'the absence of uniform gravitational settling observed throughout the IP, that is the Df/DD proportion remained nearly constant along the dust pathway across the IP (López-Cayuela et al., 2023)'*. (Page 8, lines 239-240).

13. Section 4.1: This may be a section where to introduce as well the fine and coarse mode AOD from AERONET compared to the fine and coarse mode AOD from the lidar measurements. An analysis of the internal consistency of the g for fine, coarse and total dust may be also performed. This is important for further discussion in section 4.3.

**Authors' Response:** We thank this reviewer's comment and their interesting recommendation. Nevertheless, it would shift our focus towards the differences between POLIPHON retrievals and AERONET products for dust, which is beyond the scope of this work. Our primary objective is to determine the dust radiative effect (not the aerosol radiative effect as a whole) using the lidar-derived dust extinction retrievals (e.g., from POLIPHON) and the necessary radiative properties (from AERONET) as required in the radiative transfer model simulations, including the novelty of considering two different methodologies for that purpose and examining the differences between them. POLIPHON is a widely used, robust and validated method, supported by various campaigns and studies from other authors (Ansmann et al., 2017; Mamouri and Ansmann, 2017; Córdoba-Jabonero et al., 2018, 2021; Couto et al., 2021; Salgueiro et al., 2021; Sicard et al., 2022; references included in the manuscript), giving confidence to accurately distinguish the dust fine and coarse components from other aerosols.

We did not consider using AERONET AOD values for three main reasons: first, they are columnar values, and not vertical extinction profiles as obtained from lidar measurements; second, they account not only for dust but also for background aerosols, and we were focused on the calculation of the dust radiative effect only; and third, AERONET products were available for mainly daytime (no night-time data at all the stations).

Regarding the internal consistency of $g$, we address the discussion to our previous 'Authors' response' to the reviewer's comment #9 (about 'Line 108: …'), leading to the conclusion: *'Hence, at least in this specific dust event, the assumption of a constant g value for the Df component (i.e., AERONET fine g) throughout the entire atmospheric column can be considered reliable.'* (Page 13, lines 413-414).

In addition, the corresponding text has been modified as follows:

Page 13, lines 399-414: *'As far as the fine mode is concerned, the AERONET fine g value is introduced in GAME for computing the DRE related to the Df component, as exposed in Section 2.1.1. However, it should be taken into account that the fine g is influenced by the total fine mode, i.e. both Df and background aerosols. In addition, assuming a vertically uniform g for the Df component could have substantial consequences, mainly because fine g values can be strongly*

*affected by background aerosols, which are also mostly confined to the boundary layer. Therefore, a complementary study has been conducted to study the degree of suitability of applying AERONET fine g values for DRE computation of the Df component. This way, hourly fine g values for those cases reported under dust-dominated conditions (i.e., cases where the AERONET Fine Mode Fraction is less than 40%) were compared with respect to the fine g values reported for all cases during the study period (25 March to 7 April 2021). In summary, results indicated that differences in fine g between cases under dust-dominated and all conditions were not significantly high for those obtained at each station during the study period as well as regarding the period-averaged fine g differences between stations. Specifically, the relative differences ranged between -3.4% and +0.4%. Therefore, it can be assumed that the contribution of background aerosols to the fine g can be considered negligible, and then its values can be used for the Df component. This outcome could be, to some extent, expected since background aerosols generally exhibit very low linear depolarization ratios, as they are predominantly small and spherical particles with a minimal contribution to the g parameter. Hence, at least in this specific dust event, the assumption of a constant g value for the Df component (i.e., AERONET fine g) throughout the entire atmospheric column can be considered reliable.'*

14. Section 4.2: I find this section very long and too descriptive. The benefit of the comparison with other studies is rather limited given the differences in the events, AOD, height of the dust layers, etc. I think a comparison table between studies (a probably limited to radiative efficiency) and a structure and concise discussion with some key aspect would be more informative. Consider also adding in the table estimates from dust modelling studies in the region.

**Authors' Response:** Following the Reviewer's #2 and #3 suggestions, the Section 4.2 was re-structured, and reduced by 50%. The updated text can be found in the manuscript in red (pages 10-12, lines 306-376).

15. Section 4.3: Please see my general comments to provide a comprehensive assessment of the differences between the methods. This section needs major rework and additional figures for the analysis.

**Authors' Response:** Thanks for the helpful suggestion. The Section 4.3 has been renamed ('*Differences between approaches for DRE estimation in SW radiative flux simulations*') and has been divided into two subsections ('*4.3.1 Potential limitations in the radiative properties*', and '*4.3.2 Comparative analysis*'), including the new Figures 7 and 8 by replacing the previous Figure 7, to improve the reading and present more clearly the main results. The text appears in red in the manuscript. (Pages 10-14, lines 377-446).

16. Conclusions: reconsider the conclusions in view of the new analyses performed, and emphasize much more on the implications on a broader context (for modeling and radiometric measurements).

**Authors' Response:** Please, look at the conclusions section, which has been revised according with the changes made in the manuscript and the new analysis performed. (Pages 15-16, lines 447-489).

17. What's next? How can AERONET measurements and lidar measurements be better combined? How can we assess uncertainties?

**Authors' Response:** There are other emergent methodologies (e.g., GRASP algorithm - Generalized Retrieval of Atmosphere and Surface Properties, Dubovik et al., 2014), which combines AERONET and lidar measurements to obtain vertical optical and microphysical properties of the aerosols. However, the used AERONET data in this work are constraint to provide inputs to GAME model of some radiative properties of the atmospheric aerosols. In

order to assess how to improve the input for GAME simulations by using the proposed methodology, the following text has been added (page 16, lines 483-489):

*'It is important to note that the methodology used in this work combines lidar data, which provide vertical aerosol profiles, with photometer data, which derive columnar aerosol radiative properties. One way to improve this methodology would be to use height-resolved key radiative parameters such as $\omega$ and $g$ profiles, which would be particularly useful in cases where either background aerosols play a significant role, or in scenarios involving a mixture of aerosol types. The use of multiwavelength lidars instead of single-wavelength lidar systems together with particular inversion methods could provide those required $\omega$ and $g$ profiles.'*

Reference:
Dubovik et al., 2014: GRASP: a versatile algorithm for characterizing the atmosphere. SPIE Newsroom, doi:10.1117/2.1201408.005558.

18. Figure2: improve color scale. We cannot see the low values with this space.
**Authors' Response:** The colour scale of the Figure 2 has been improved (see new Fig. 2 at the end).
* * *
**Reviewer #3's comments**
The study presents the direct radiative effect of dust particles in the shortwave range over an extended dust intrusion event over the Iberian Peninsula. The study is interesting, in particular regarding the comparisons between a novel approach that quantity size-dependent DRE versus a commonly used approach that quantifies DRE based on total dust. The manuscript is well written, however a bit difficult to follow with all the different abbreviations. The manuscript requires some major revisions in the discussion and presentation of key findings before it can be published in ACP.

General comments:
1. Do and by how much the AERONET vs lidar AODs differ in terms of fine, coarse, and total dust aerosols? How would the differences between these two networks affect the conclusion of this study?
**Authors' response:** We thank this reviewer's comment and their interesting recommendation. Nevertheless, it would shift our focus towards the differences between POLIPHON retrievals and AERONET products for dust, which is beyond the scope of this work. Our primary objective is to determine the dust radiative effect (not the aerosol radiative effect as a whole) using the lidar-derived dust extinction retrievals (e.g., from POLIPHON) and the necessary radiative properties (from AERONET) as required in the radiative transfer model simulations, including the novelty of considering two different methodologies for that purpose and examining the differences between them. POLIPHON is a widely used, robust and validated method, supported by various campaigns and studies from other authors (see references included in the manuscript), giving confidence to accurately discriminate the dust fine and coarse components from other aerosols.

We did not consider using AERONET AOD values for three main reasons: first, they are columnar values, and not vertical extinction profiles as obtained from lidar measurements; second, they account for not only dust but also background aerosols, and we were focused on the calculation of the dust radiative effect only; and third, AERONET products were available for mainly daytime (no night-time data at all the stations).

2. More critically, it is very descriptive in the presentation of the results, i.e. it presents the results in the tables and figures but does not discuss the potential reasons of these differences, e.g. between stations, between AERONET and Lidars, and especially between DRE calculation approaches, which is the main selling part of the study.

**Authors's Response:** Following the Reviewer's considerations, the sections 4.1, 4.2 and 4.3 has been re-written (pages 9-14), and the previous Figure 7 has been replaced by the new Figures 7 and 8. The section 4.1 shows an analysis of the specific radiative dust properties introduced in the model. The section 4.2 is devoted to describing the episode in terms of the evolution of the dust properties and DRE across the stations considered. The section 4.3 assess the differences between approaches for DRE estimation in SW radiative flux simulations. As it is a common concern of several Reviewers, the text can be found in red in the manuscript.

3. Although it is very good that the findings are compared with previous estimates but is difficult to follow. Maybe a table could help to improve this?

**Authors' Response:** The authors appreciate the reviewer's suggestion. In response to the recommendation of several reviewers, instead of a table, the text has been modified accordingly (Sect. 4.2, pages 10-12, lines 306-376). As it is a common concern of several Reviewers, the text can be found in red in the manuscript.

Specific comments:
4. Ignoring the fine fraction in DRE calculations in the previous statements is a strong statement and is misleading.

**Authors' Response:** To be more accurate, the original sentence: '*in which the potential impact of Df particles on the total radiative effect has been mostly ignored',* has been changed to: '*in which the potential impact of Df particles on the total radiative effect has not been separately underlined*'. (Page 3, lines 78-79).

5. Line 108: What are the implications of having vertically constant $g$ values, in particular in the BL or over clean vs polluted background sites?

**Authors' Response:** As previously responded to reviewer #2's comment #9, introducing vertically constant $g$ values, as derived from the columnar AERONET products, into the model can have a notable impact, particularly for the fine mode, due to the additional presence of background (fine) aerosols at the boundary layer. However, in this specific event, the contribution of these background aerosols to the fine-mode $g$ values is considered negligible, as explained in the paragraph below. Therefore, this assumption does not significantly affect the results obtained.

An additional analysis was conducted to support such a statement, with the key findings presented in Section 4.3, which has been divided into two subsections for clarity. The text has been revised accordingly (page 13, lines 399-414):

*'As far as the fine mode is concerned, the AERONET fine $g$ value is introduced in GAME for computing the DRE related to the Df component, as exposed in Section 2.1.1. However, it should be taken into account that the fine $g$ is influenced by the total fine mode, i.e. both Df and background aerosols. In addition, assuming a vertically uniform $g$ for the Df component could have substantial consequences, mainly because fine $g$ values can be strongly affected by background aerosols, which are also mostly confined to the boundary layer. Therefore, a complementary study has been conducted to study the degree of suitability of applying AERONET fine $g$ values for DRE computation of the Df component. This way, hourly fine $g$ values for those cases reported under dust-dominated conditions (i.e., cases where the AERONET Fine Mode Fraction is less than 40%) were compared with respect to the fine $g$ values reported for all cases during the study period (25 March to 7 April 2021). In summary, results indicated that differences in fine $g$ between cases under dust-dominated and all conditions were not significantly high for*

*those obtained at each station during the study period as well as regarding the period-averaged fine g differences between stations. Specifically, the relative differences ranged between -3.4% and +0.4%. Therefore, it can be assumed that the contribution of background aerosols to the fine g can be considered negligible, and then its values can be used for the Df component. This outcome could be, to some extent, expected since background aerosols generally exhibit very low linear depolarization ratios, as they are predominantly small and spherical particles with a minimal contribution to the g parameter. Hence, at least in this specific dust event, the assumption of a constant g value for the Df component (i.e., AERONET fine g) throughout the entire atmospheric column can be considered reliable.'*

As this is a concern shared by other Reviewers, the text is highlighted in the manuscript in red.

6. Line 114: Can you explain how the dust extinction coefficients from POLIPHON are degraded?

**Authors' Response:** The lidar-derived dust extinction profiles have different vertical resolutions (7.5 m, 15 m, and 75 m) depending on the lidar system used. To homogenize all the data, these extinction profiles have been degraded to the 18 layers of the GAME model through trapezoidal numerical integration. The corresponding explanation has been added (page 5, lines 146-148)*: 'The coarse and fine lidar-derived $\alpha^{532}$ profiles were previously obtained in López-Cayuela et al. (2023) (see Sect. 3). As those profiles have different vertical resolutions depending on the lidar system used, they have been degraded to the 18 model layers (ranging from the surface up to 20 km height) through trapezoidal numerical integration in order to homogenize all the datasets.'*

7. Line 176: This definition is not used for aging. This is simply gravitational settling as described later in the sentence.

**Authors' Response:** Thanks for the comment. The sentence *'the absence of dust ageing observed throughout the IP'* has been changed to: *'the absence of uniform gravitational settling observed throughout the IP, that is the Df/DD proportion remained nearly constant along the dust pathway across the IP (López-Cayuela et al., 2023),'* (page 8, lines 239-240).
* * *
**Reviewer #4's comments**
This study quantifies the dust direct radiative effect (DRE) in the short-wave range (SW) during a prolonged dust episode over the Iberian Peninsula. The analyses were performed over five lidar stations. The study uses two distinct methodologies to simulate the SW DRE. One separately estimates the effect of fine and coarse dust particles, and one estimates the effect of the total dust. The study highlights that the fine fraction cannot be disregarded as it contributes to nearly half of the total DRE at the top of the atmosphere (TOA) and bottom of the atmosphere (BOA). The differences between the two methodologies are attributed to differences in the asymmetry factor between the fine mode and total dust components.

The manuscript requires some major revision before it can be published in ACP.

General Comments:
1. This study points out the comparison of the two different methodologies as its main highlight. The study is interesting and has potential implications for evaluating size-dependent DRE. The differences between the two methodologies are not explored in detail. While the planned future studies with dust episodes that exhibit higher variability in the fine-to-coarse ratio would provide additional insight, some considerations can be discussed more within this study.

**Authors's Response:** Regarding this reviewer's comment and suggestions, the Section 4.3 (Differences between approaches for DRE estimation in SW radiative flux simulations, pages 12-14, lines 377-446) has been rewritten to perform a deeper analysis. As this is a common concern of the other Reviewers, this section can be found in red in the manuscript.

2. What could be worth considering are the differences in AERONET and POLIPHON fine-to-coarse ratios and their impact on the DRE estimates. Additionally, a comment on the other aerosols present above the stations, particularly the boundary layer aerosols and their possible contribution.

**Authors' Response:** We thank this reviewer's comment and their interesting recommendation. Nevertheless, it would shift our focus towards the differences between POLIPHON retrievals and AERONET products for dust, which is beyond the scope of this work. Our primary objective is to determine the dust radiative effect (not the aerosol radiative effect as a whole) using the lidar-derived dust extinction retrievals (e.g., from POLIPHON) and the necessary radiative properties (from AERONET) as required in the radiative transfer model simulations, including the novelty of considering two different methodologies for that purpose and examining the differences between them. POLIPHON is a widely used, robust and validated method, supported by various campaigns and studies from other authors (see references included in the manuscript), giving confidence to accurately discriminate the dust fine and coarse components from other aerosols.

We did not consider using AERONET AOD values for three main reasons: first, they are columnar values, and not vertical extinction profiles as obtained from lidar measurements; second, they account for not only dust but also background aerosols, and we were focused on the calculation of the dust radiative effect only; and third, AERONET products were available for mainly daytime (no night-time data at all the stations).

In addition, information about the aerosols present at the boundary layer in the stations has been added (pages 7-8, lines 218-228):

*'All these stations share the commonality of being dedicated to aerosol-cloud monitoring. Among several instrumentation, each station is equipped with an AERONET photometer (or is close to an AERONET station) and a lidar system. Those five stations share a common exposure to Saharan dust outbreaks, particularly during spring and summer months, albeit with varying frequencies (i.e., Córdoba-Jabonero et al., 2021; López-Cayuela et al., 2023). Moreover, each station exhibits a unique aerosol background. Particularly, the aerosol background at the ARN station is mostly from marine and rural origin, as ARN is placed in a rural environment at the southwestern IP, and less than 1 km from the Atlantic coastline. EVO station is located in a rural region with limited industrialization and low levels of anthropogenic aerosol concentrations (Pereira et al., 2009; Preissler et al., 2013). Both GRA and TRJ stations are located in populated cities, and their background aerosols are of anthropogenic origin (Lyamani et al., 2012; Molero et al., 2014). Finally, BCN station is located on the northeast coast of the IP, within a densely populated and industrialized region, being thus the background aerosol load predominantly composed by urban and marine aerosols (Sicard et al., 2011).'*

References added:

- Lyamani, H., Fernández-Gálvez, J., Pérez-Ramírez, D., Valenzuela, A., Antón, M., Alados, I., Titos, G., Olmo, F. J., and Alados-Arboledas, L.: Aerosol properties over two urban sites in South Spain during an extended stagnation episode in winter season, Atmos. Environ., 62, 424-432, doi:10.1016/j.atmosenv.2012.08.050, 2012.
- Molero, F., Andrey, F. J., Fernandez, A. J., Parrondo, M. D. C., Pujadas, M., Córdoba-Jabonero, C., Revuelta, M. A., and Gomez-Moreno, F. J.: Study of vertically resolved aerosol properties over an urban background site in Madrid (Spain), *Int. J. Remote Sens., 35*(6), 2311-2326, doi:10.1080/01431161.2014.894664, 2014.

- Pereira, S. N., Wagner, F., and Silva, A. M.: Continuous measurements of near surface aerosols in the south-western European (Portugal) region in 2006–2008, Adv. Sci. Res., *3*(1), 1-4, doi:10.5194/asr-3-1-2009, 2009.
- Preißler, J., Wagner, F., Guerrero-Rascado, J. L., and Silva, A. M.: Two years of free-tropospheric aerosol layers observed over Portugal by lidar, J. Geophys. Res. Atmos., *118*(9), 3676-3686, doi:10.1002/jgrd.50350, 2013.
- Sicard, M., Rocadenbosch, F., Reba, M. N. M., Comerón, A., Tomás, S., García-Vízcaino, D., Batet, O, Barrios, R., Kumar, D., and Baldasano, J. M.: Seasonal variability of aerosol optical properties observed by means of a Raman lidar at an EARLINET site over Northeastern Spain, *Atmos. Chem. Phys.*, *11*(1), 175-190, doi:10.5194/acp-11-175-2011, 2011.

3. How do the uncertainties of the POLIPHON retrieval affect the fine-to-coarse ratio and therefore the differences in the two DRE estimation methodologies?

**Authors's Response:** POLIPHON uncertainties for the total, coarse and fine dust extinction coefficents are 15%-25%, 20%-30% and 30%-50%, respectively (Ansmann et al., 2019). Thus, it is expected that the DRE[(I)] uncertainty is higher than that DRE[(II)] one. This consideration has been included in the text:

Page 5, lines 143-145: 'The uncertainties in the calculation of the POLIPHON $\alpha^{532}$ are 30%-50%, 20%-30%, and 15%-25%, for Df, Dc and total dust, respectively (Ansmann et al., 2019).'

Page 7, lines 201-203: *'Moreover, it should be considered that the uncertainties in DRE(I) should be higher than in DRE(II), as derived from the uncertainties in the calculation of the POLIPHON $\alpha$^532, which are higher for Df and Dc modes than for total dust (see Sect. 2.1.2).'*

Specific Comments:
4. Line 48: The way this is phrased does not clarify the association between dust intrusions and heat waves. Is it that the the particular synoptic conditions are favourable both for the intrusion and the heatwave?

**Authors' Response:** Indeed, certain weather pattern favour Saharan desert outbreaks over the Iberian Peninsula. In order to explain this fact, and respond to this reviewer's comment, new references that support the relationship of the heatwaves and part of the dust intrusions has been added. The text has been also modified as follows (page 2, lines 39-53):

*'On one hand, the geographical proximity of the Iberian Peninsula (IP) to North Africa, as well as the persistence of certain favourable weather patterns (Russo et al., 2020; Couto et al., 2021), make the IP one of the main pathway regions for Saharan desert dust transport towards Europe. On the other hand, Sánchez-Benítez et al. (2020) found that Iberian heatwave events are primarily linked to anomalous atmospheric circulation, typically characterized by a pronounced positive 500 hPa geopotential height anomaly (Z500) aloft. In particular, an analysis of daily weather regimes during Iberian heatwave days reveals a marked dominance of positive Z500 anomalies over western Europe, resembling the occurrence of Euro-Atlantic subtropical ridges. The frequency of this occurrence during heatwave days is observed to double relative to climatological averages. Last studies linked both processes, showing robust evidence for increases in maximum temperatures and the frequency of heatwaves over Europe (IPCC, 2023), which can be partially associated with dust intrusions (Sousa et al., 2019; Fernandes and Fragoso, 2021; Barriopedro et al., 2023).'*

As this is a concern shared by other Reviewers, the text is highlighted in the manuscript in red.

References added:

- Barriopedro, D., García-Herrera, R., Ordóñez, C., Miralles, D. G., and Salcedo-Sanz, S.: Heat waves: Physical understanding and scientific challenges, Rev. Geophys., 61, e2022RG000780, doi:10.1029/2022RG000780, 2023.

- Sánchez-Benítez, A., Barriopedro, D., García-Herrera, R.:Tracking Iberian heatwaves from a new perspective, Weather and Climate Extremes, 28, 100238, doi:j.wace.2019.100238, 2020.

5. Line 71: The statement that the fine particles have been ignored in previous studies should be referenced. Additionally, in the following sentences, it is stated that they have not been explicitly ignored but considered as a part of the total dust. Perhaps what was meant was that the contribution of the fine particles separately has not been researched as extensively as in this study.

**Authors' Response:** To be more accurate, the original sentence: '*in which the potential impact of Df particles on the total radiative effect has been mostly ignored',* has been changed to: '*in which the potential impact of Df particles on the total radiative effect has not been separately underlined.*' (page 3, lines 78-79). As this is a concern shared by other Reviewers, the text is highlighted in the manuscript in red.

6. Line 103: What is the impact of using the AERONET retrieved asymmetry parameter, particularly as related to the boundary layer aerosols?

**Authors' Response:** As previously responded to reviewer #2's comment #9 and reviewer #3's comment #5, introducing vertically constant $g$ values, as derived from the columnar AERONET products, into the model can have a notable impact, particularly for the fine mode, due to the additional presence of background (fine) aerosols at the boundary layer. However, in this specific event, the contribution of these background aerosols to the fine-mode $g$ values is considered negligible, as explained in the paragraph below. Therefore, this assumption does not significantly affect the results obtained.

An additional analysis was conducted to support such a statement, with the key findings presented in Section 4.3, which has been divided into two subsections for clarity. The text has been revised accordingly (page 13, lines 399-414):

'*As far as the fine mode is concerned, the AERONET fine $g$ value is introduced in GAME for computing the DRE related to the Df component, as exposed in Section 2.1.1. However, it should be taken into account that the fine $g$ is influenced by the total fine mode, i.e. both Df and background aerosols. In addition, assuming a vertically uniform $g$ for the Df component could have substantial consequences, mainly because fine $g$ values can be strongly affected by background aerosols, which are also mostly confined to the boundary layer. Therefore, a complementary study has been conducted to study the degree of suitability of applying AERONET fine $g$ values for DRE computation of the Df component. This way, hourly fine $g$ values for those cases reported under dust-dominated conditions (i.e., cases where the AERONET Fine Mode Fraction is less than 40%) were compared with respect to the fine $g$ values reported for all cases during the study period (25 March to 7 April 2021). In summary, results indicated that differences in fine $g$ between cases under dust-dominated and all conditions were not significantly high for those obtained at each station during the study period as well as regarding the period-averaged fine $g$ differences between stations. Specifically, the relative differences ranged between -3.4% and +0.4%. Therefore, it can be assumed that the contribution of background aerosols to the fine $g$ can be considered negligible, and then its values can be used for the Df component. This outcome could be, to some extent, expected since background aerosols generally exhibit very low linear depolarization ratios, as they are predominantly small and spherical particles with a minimal contribution to the $g$ parameter. Hence, at least in this specific dust event, the assumption of a constant $g$ value for the Df component (i.e., AERONET fine $g$) throughout the entire atmospheric column can be considered reliable.*'

As this is a concern shared by other Reviewers, the text is highlighted in the manuscript in red.

[Figure]

*Figure 2: Temporal evolution of the DD extinction coefficient ($\alpha_{DD}^{532}$, km$^{-1}$) at the five Iberian lidar stations (from North-East to South-West IP, by decreasing latitude): a) Barcelona (BCN), b) Torrejón/Madrid (TRJ), c) Évora (EVO), d) Granada (GRA) and e) El Arenosillo/Huelva (ARN). Profile gaps correspond to either no inversion available or no lidar measurements.*

[Figure]

**Figure 7. Relative differences in DRE ($\Delta^{rel}DRE$; %) as obtained between the two approaches as a function of SZA at: a) BOA and b) TOA for all the five lidar stations involved in this study, from 25 March to 7 April 2021. The dependence on DD $DOD^{532}$ is shown by a colour-scaled bar. The shadowed box marks the threshold of $\Delta^{rel}DRE$ = ±15%.**

[Figure]

**Figure 8. Differences in DRE ($\Delta DRE$; W m$^{-2}$) as obtained between the two approaches (DRE$^{(I)}$ - DRE$^{(II)}$; see Eq. 6) at: a) BOA, and b) TOA for all the five lidar stations involved in this study, from 25 March to 7 April 2021. The dependence on the DD $DOD^{532}$ is shown by a colour-scaled bar. The profiles with SZA > 70° are marked with a cross.**

*Table 1: Input parameters and radiative properties for the GAME model and databases in the SW spectral range. Note that $z$ denotes the vertical dependence. DRE$^{(I)}$ and DRE$^{(II)}$ denotes the particular approach used for DRE simulation, considering either dust component separation (Dc, Df) or total dust (DD, no separation), respectively.*

| | Parameters | | Database / instrumentation |
|---|---|---|---|
| | DRE$^{(I)}$ | DRE$^{(II)}$ | |
| Atmosphere and Land | SA | | MODIS |
| | Atmospheric profiles | | U.S. std. atm. + 3h GDAS profiles |
| | Gas concentration profiles | | U.S. std. atm. + 3h GDAS profiles |
| | Absorption coefficients | | HITRAN |
| Aerosols | Df or Dc $\alpha^{532}(z)$ | DD $\alpha^{532}(z)$ | Lidar |
| | Df or Dc DOD | DD DOD | Lidar |
| | fine or coarse $g$ | total $g$ | Sun-photometer |
| | total $\omega$ | | Sun-photometer |
| | $AE^{440-870}$ | | Sun-photometer |